# Tortoise and Hare Guidance: Accelerating Diffusion Model Inference with Multirate Integration

**Yunghee Lee**     **Byeonghyun Pak**     **Junwha Hong**     **Hoseong Kim**[†]

Agency for Defense Development

{yhl, byeonghyun_pak, qbit, hoseongkim}@add.re.kr

## Abstract

In this paper, we propose **Tortoise and Hare Guidance (THG)**, a training-free strategy that accelerates diffusion sampling while maintaining high-fidelity generation. We demonstrate that the noise estimate and the additional guidance term exhibit markedly different sensitivity to numerical error by reformulating the classifier-free guidance (CFG) ODE as a *multirate system of ODEs*. Our error-bound analysis shows that the additional guidance branch is more robust to approximation, revealing substantial redundancy that conventional solvers fail to exploit. Building on this insight, THG significantly reduces the computation of the additional guidance: the noise estimate is integrated with the tortoise equation on the original, fine-grained timestep grid, while the additional guidance is integrated with the hare equation only on a coarse grid. We also introduce (i) an error-bound-aware timestep sampler that adaptively selects step sizes and (ii) a guidance-scale scheduler that stabilizes large extrapolation spans. THG reduces the number of function evaluations (NFE) by up to 30% with virtually no loss in generation fidelity ($\Delta$ImageReward $\leq 0.032$) and outperforms state-of-the-art CFG-based training-free accelerators under identical computation budgets. Our findings highlight the potential of multirate formulations for diffusion solvers, paving the way for real-time high-quality image synthesis without any model retraining. The source code is available at `https://github.com/yhlee-add/THG`.

## 1   Introduction

Diffusion models (DMs) have become the state-of-the-art generative model for images [10, 40, 47] and, more recently, for video [20, 1, 52, 21] and audio-visual content [5, 41]. Despite their impressive quality, sampling is costly: each output is obtained by iteratively denoising a noisy sample, and the latency scales with the total number of function evaluations (NFE) required by the solver.

Many practical scenarios, such as text-to-image synthesis, class-controlled synthesis, or in-context image editing, require conditional generation. The dominant technique for high-quality conditioning is *classifier-free guidance* (CFG) [18], which improves perceptual quality and controllability. However, CFG runs the denoising network twice per timestep—once conditional and once unconditional—thereby doubling the NFE. For real-time applications, such as interactive editing and large-scale serving, evaluating a deep backbone at every timestep remains a major bottleneck.

A large body of work to accelerate these models has focused on two main approaches. Some approaches reduce the number of steps using higher-order ODE/SDE solvers [45, 46, 30] or distillation [43, 34], while others—such as cache-based strategies like DeepCache [33] and Learning-to-Cache [32]—lower the cost per step by reusing intermediate features. Nevertheless, both approaches still perform two forward passes whenever CFG is enabled, implicitly assuming that conditional and unconditional calls are equally indispensable.

---

[†]Corresponding author.

39th Conference on Neural Information Processing Systems (NeurIPS 2025).

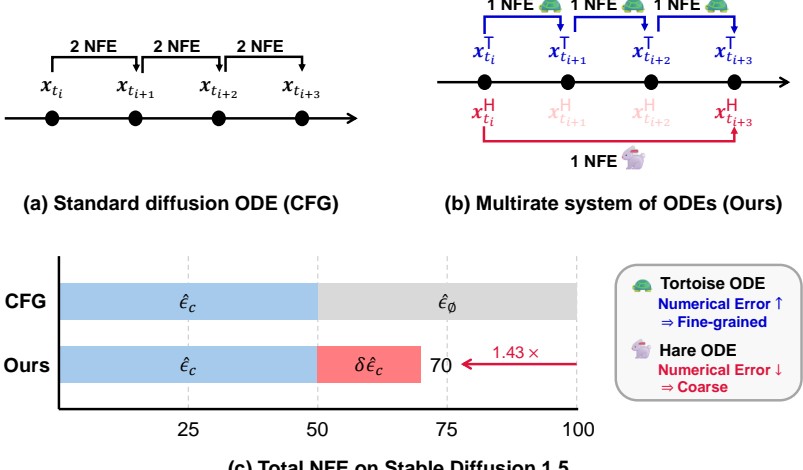

**Figure 1: Conceptual illustration of Tortoise and Hare Guidance.** We decompose the standard diffusion ODE into a tortoise branch (Eq. 6), which is numerically sensitive and thus integrated on a fine-grained grid, and a hare branch (Eq. 7), which is comparatively less sensitive and can be integrated with larger step sizes. Our multirate scheme evaluates each branch at different timestep grids, skipping unnecessary evaluations, thereby boosting inference efficiency without sacrificing sample quality.

Through the lens of numerical analysis, we revisit CFG by reformulating the reverse diffusion process as a two-state multirate system of ODEs whose trajectories are governed by the noise estimate and the additional guidance term. Our error-bound analysis reveals a pronounced asymmetry: the additional guidance term is more robust to approximation than the noise estimate, exposing substantial redundancy that conventional solvers fail to exploit. This finding raises a natural question: *Do we need to compute the neural network twice at every fine-grained timestep?*

Leveraging this asymmetry, we introduce **Tortoise and Hare Guidance (THG)**, a training-free sampler that bypasses most additional guidance computation. The noise estimate is integrated with the tortoise equation on the original fine-grained timestep grid. Meanwhile, the additional guidance is integrated with the hare equation only on a coarse grid. We further introduce (i) an error-bound-aware timestep sampler that adaptively determines the coarse grid, and (ii) a guidance-scale scheduler that keeps the trajectory stable over significant gaps.

With these components, THG achieves sampling speeds up to $1.43\times$ faster by reducing the NFE budget from 100 to as low as 70 while maintaining virtually identical generation fidelity ($\Delta$ImageReward $\leq 0.032$). Moreover, across Stable Diffusion 1.5 [40], 3.5 Large [47], and Audio-LDM 2 [28], our method outperforms state-of-the-art CFG-based training-free accelerators under identical computation budgets. Our study highlights the potential of multirate formulations for accelerating diffusion models and brings us a step closer to achieving real-time performance and high-quality image synthesis without retraining the model.

In summary, our contributions are threefold:

- We are the first to cast the reverse diffusion ODE as a two-state multirate system of ODEs and to provide an error-bound analysis showing that the additional guidance term can be safely approximated at a much coarser temporal resolution.

- We design **Tortoise and Hare Guidance (THG)**, a training-free sampler that eliminates the need for a significant amount of additional guidance term evaluation. THG is compatible with any diffusion backbone.

- Using image-text pairs from the COCO 2014 dataset, we demonstrate that THG can reduce NFEs up to 30% with virtually no loss in generation fidelity ($\Delta$ImageReward $\leq 0.032$). THG outperforms state-of-the-art CFG-based accelerators under identical compute budgets.

## 2 Related work

**Diffusion models**  Denoising Diffusion Probabilistic Models (DDPMs) [19] laid the foundation for modern diffusion models by introducing a probabilistic framework. A forward Markov process gradually corrupts a data point $x_0$ into Gaussian noise. In the reverse process, at each timestep $t$, a neural network $\hat{\epsilon}_\theta(x_t, t)$ estimates and removes the noise component in $x_t$ to recover $x_{t-1}$, ultimately reconstructing $x_0$. The denoising trajectory can be interpreted either as a stochastic differential equation (SDE) or its deterministic counterpart, the probability flow ODE (PF-ODE) [46]. Denoising Diffusion Implicit Models (DDIMs) [45] drop the strict Markov assumption of DDPMs and apply Tweedie's formula [9] to jump directly from $x_t$ to $x_s$, cutting sampling steps from hundreds of steps to as few as 50 and effectively solving the PF-ODE in a single deterministic pass [46].

**ODE-based integrators**  Viewing diffusion sampling as an initial-value ODE problem enables high-order integration techniques. Concretely, DPM-solver [30] observes that the diffusion ODE

$$\mathrm{d}x_t/\mathrm{d}t = f(t)x_t + (g^2(t)/2\sigma_t)\hat{\epsilon}_\theta(x_t) \tag{1}$$

has a semi-linear term $f(t)x_t$. The need for approximation for the linear term is eliminated by solving the semi-linear ODE using the *variation of constants* formula. This semi-linear integrator then affords large step sizes with minimal approximation error. Inspired by these semi-linear methods, we introduce a multirate formulation for the classifier-free guidance (CFG) scheme [18] that adjusts the step size of each component of CFG to its own dynamics, achieving further reductions in the number of function evaluations (NFE) without degrading sample quality.

**Classifier-free guidance and its variations**  In real-world applications, diffusion models must produce samples that satisfy a given condition (e.g., class label or text prompt). Classifier Guidance [8] achieves this by incorporating a pre-trained classifier $p_\phi(c|x_t)$, effectively sampling from the *sharpened* density $p(x)p(c|x)^\omega$, where $\omega$ controls the strength of the bias towards class $c$. Classifier-Free Guidance (CFG) [18] eliminates the need for an external classifier by training a single denoising network that gives both conditional and unconditional outputs. Concretely, if $\hat{\epsilon}_\theta(x_t, c)$ and $\hat{\epsilon}_\theta(x_t, \varnothing)$ denote the network's noise predictions with and without condition $c$, respectively, then CFG defines

$$\hat{\epsilon}_\theta^{\mathrm{CFG}}(x_t, c) = \hat{\epsilon}_\theta(x_t, \varnothing) + \omega \cdot (\hat{\epsilon}_\theta(x_t, c) - \hat{\epsilon}_\theta(x_t, \varnothing)). \tag{2}$$

Subsequent variants focus on finding the optimal strength and timing of guidance for balancing condition fidelity against sample diversity. Guidance Interval [26] restricts the use of CFG to mid-level noise steps, avoiding over-conditioning at the beginning and final stages of the sampling process. CADS and Dynamic-CFG [42] slowly anneal either the conditioning vector or the scale $\omega$ during the early denoising steps, preserving diversity in the final samples. PCG [2] reformulates CFG as a predictor-corrector method (with $\omega' = 2\omega - 1$) that alternates between denoising and sharpening phases. CFG++ [7] treats guidance as an explicit loss term rather than a sampling bias, splitting each DDIM iteration into "denoising" and "renoising" phases. Unlike these methods, we reformulate the diffusion ODE using a multirate method, integrating the noise estimate on a fine-grained grid and the additional guidance term on a coarse grid, reducing the NFE while preserving sample quality.

**Efficient diffusion models**  Beyond advanced ODE/SDE solvers, various methods have been proposed to speed up pre-trained diffusion models. Distillation methods [43, 34] compress a pre-trained "teacher" model into a "student" model that can advance multiple timesteps in one forward pass. While these methods reduce the number of sampling steps, they incur substantial retraining costs. Cache-based techniques exploit feature redundancy within the denoising neural network $\hat{\epsilon}_\theta$. DeepCache [33] reuses high-level U-Net activations across adjacent steps. Learning-to-Cache [32] introduces a layer-wise caching mechanism that dynamically reuses transformer activations across timesteps via a timestep-conditioned router. $\Delta$-Dit [4] leverages stage-adaptive caching of block-specific feature offsets in DiT models to speed up inference without retraining. These methods deliver inference speedups without retraining but depend heavily on the model's internal architecture. More recently, several works have noted that CFG doubles the NFE per denoising step and have proposed methods to reduce this extra cost. Adaptive Guidance [3] adaptively skips redundant guidance steps based on cosine similarity between conditional and unconditional predictions. FasterCache [31] reuses attention features and conditional-unconditional residuals to mitigate CFG overhead. Although these methods reduce the NFE, they lack a rigorous theoretical foundation and leave further savings on the table. Our approach delivers a more efficient and theoretically grounded method of guided diffusion by directly exploiting the CFG's intrinsic dynamics.

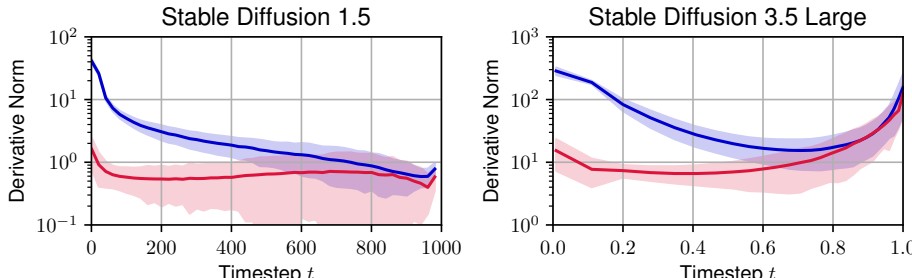

Figure 2: **Time-derivative norms of the noise estimate $\hat{\epsilon}_c(x_t)$ and additional guidance $\Delta\hat{\epsilon}_c(x_t)$.** We plot the L2 norms of the time derivatives $\frac{d}{dt}\hat{\epsilon}_c(x_t)$ and $\frac{d}{dt}\Delta\hat{\epsilon}_c(x_t)$ across diffusion timesteps for Stable Diffusion 1.5 and 3.5 Large. The results confirm that the noise estimate exhibits greater temporal sensitivity compared to the guidance term. Shaded areas denote two standard deviations over multiple prompts.

## 3 Method

In this section, we introduce **Tortoise and Hare Guidance (THG)**, which accelerates diffusion model inference by leveraging the asymmetry between the noise estimate and the additional guidance terms. Since the additional guidance term varies more slowly *w.r.t.* the denoising timestep $t$ than the noise estimate term, we apply a multirate integration scheme that uses a coarser timestep grid for the additional guidance term (Sec. 3.1 and Sec. 3.2). We then perform an approximation error-bound analysis to determine the appropriate grid granularity (Sec. 3.3). Finally, we propose an adaptive guidance scale to compensate for any performance degradation resulting from the reduced number of evaluation points (Sec. 3.4).

**Preliminaries** To accommodate different definitions of the diffusion process [19, 46, 49], we adopt a general notation [30] so that the forward process and the diffusion ODE are described as follows:

$$q(x_t|x_0) := \mathcal{N}(x_t; \alpha_t x_0, \sigma_t^2 I), \quad \frac{dx_t}{dt} = f(t)x_t + \frac{g^2(t)}{2\sigma_t}\hat{\epsilon}_\theta(x_t), \quad x_T \sim \mathcal{N}(0, \sigma_T^2 I), \quad (3)$$

where $f(t) = \frac{d\log\alpha_t}{dt}$, $g^2(t) = \frac{d\sigma_t^2}{dt} - 2\frac{d\log\alpha_t}{dt}\sigma_t^2$, and $t \in [0, T]$. ($v$-prediction models are covered in Appendix A.) $\alpha_t$ and $\sigma_t$ are the predefined noise schedule of the diffusion model. Although modern diffusion models primarily operate in the latent space [40], we adopt $x$ (instead of $z$), as our framework is agnostic to this choice. For brevity, we denote the unconditional noise estimate $\hat{\epsilon}_\varnothing(x_t) := \hat{\epsilon}_\theta(x_t, \varnothing)$, the conditional noise estimate $\hat{\epsilon}_c(x_t) = \hat{\epsilon}_\theta(x_t, c)$, the difference of the two $\Delta\hat{\epsilon}_c(x_t) := \hat{\epsilon}_c(x_t) - \hat{\epsilon}_\varnothing(x_t)$, and the CFG noise estimate $\hat{\epsilon}_c^\omega(x_t) = \hat{\epsilon}_\theta^{\text{CFG}}(x_t, c)$ following [7].

### 3.1 A multirate formulation

We propose a multirate formulation [39], in which the reverse diffusion process is decomposed into numerically sensitive and less sensitive components to reduce the number of function evaluations (NFE). We begin by writing the diffusion ODE in Eq. 3 by explicitly separating it into two distinct terms, the noise estimate and the additional guidance term. By the definition of CFG, we have

$$\hat{\epsilon}_\theta(x_t) := \hat{\epsilon}_c^\omega(x_t) = \hat{\epsilon}_\varnothing(x_t) + \omega \cdot \Delta\hat{\epsilon}_c(x_t) \equiv \hat{\epsilon}_c(x_t) + (\omega - 1) \cdot \Delta\hat{\epsilon}_c(x_t). \quad (4)$$

Substituting Eq. 4 into Eq. 3 yields the following:

$$\frac{d}{dt}x_t = f(t)x_t + \frac{g^2(t)}{2\sigma_t}\hat{\epsilon}_c^\omega(x_t) = f(t)x_t + \underbrace{\frac{g^2(t)}{2\sigma_t}\hat{\epsilon}_c(x_t)}_{\text{sensitive}} + \underbrace{\frac{g^2(t)}{2\sigma_t}(\omega - 1)\Delta\hat{\epsilon}_c(x_t)}_{\text{less sensitive}}. \quad (5)$$

We observe a significant difference in temporal sensitivity between the noise estimate term and the additional guidance term. Figure 2 plots the time-derivative norms of $\hat{\epsilon}_c(x_t)$ and $\delta\hat{\epsilon}_c(x_t)$, confirming that the noise estimate varies more rapidly than the additional guidance term. This result

---

**Algorithm 1** Tortoise and Hare Guidance Algorithm

---

**Require:** $x_T \sim \mathcal{N}(0, \sigma_T^2 I)$          ▷ Initial noise
**Require:** $\omega \geq 0$          ▷ Guidance scale
**Require:** $\{t_i\}_{0 \leq i \leq N}, t_0 = T, t_N = 0$          ▷ Fine-grained timestep grid
**Require:** $C \subset \{t_i | 0 \leq i \leq N\}, 0 \in C, T \in C$          ▷ Coarse timestep grid
1:   $x_T^{\mathsf{T}} \leftarrow x_T$
2:   $x_T^{\mathsf{H}} \leftarrow 0$
3:   **for** $i = 0$ **to** $N - 1$ **do**
4:      $\hat{\epsilon}_c \leftarrow \hat{\epsilon}_\theta(x_{t_i}^{\mathsf{T}} + x_{t_i}^{\mathsf{H}}, c)$          ▷ 1 NFE
5:      $x_{t_{i+1}}^{\mathsf{T}} \leftarrow \text{Solver}(x_{t_i}^{\mathsf{T}}, \hat{\epsilon}_c, t_i, t_{i+1})$          ▷ Compute $x_{t_{i+1}}^{\mathsf{T}}$ given $x_{t_i}^{\mathsf{T}}$
6:      **if** $t_i \in C$ **then**
7:          $\hat{\epsilon}_\varnothing \leftarrow \hat{\epsilon}_\theta(x_{t_i}^{\mathsf{T}} + x_{t_i}^{\mathsf{H}}, \varnothing)$          ▷ 1 NFE (only if $t_i \in C$)
8:          $\Delta\hat{\epsilon}_c \leftarrow \hat{\epsilon}_c - \hat{\epsilon}_\varnothing$
9:          $j \leftarrow i$
10:         **repeat**          ▷ Compute $x^{\mathsf{H}}$ up to the next coarse timestep
11:             $j \leftarrow j + 1$
12:             $x_{t_j}^{\mathsf{H}} \leftarrow \text{Solver}(x_{t_i}^{\mathsf{H}}, (\omega - 1) \cdot \Delta\hat{\epsilon}_c, t_i, t_j)$          ▷ Compute $x_{t_j}^{\mathsf{H}}$ given $x_{t_i}^{\mathsf{H}}$
13:         **until** $t_j \in C$          ▷ $t_j$ equals the next coarse timestep at inner loop exit
14:      **end if**
15: **end for**
16: $x_0 \leftarrow x_0^{\mathsf{T}} + x_0^{\mathsf{H}}$
17: **return** $x_0$

---

clearly demonstrates that the noise estimate exhibits greater numerical sensitivity than the additional guidance.

This motivates the use of a multirate method [44] where the sensitive term is integrated on a fine-grained grid, and the less sensitive term is integrated on a coarse grid. We split the diffusion ODE (Eq. 5) into the following system of ODEs:

$$\frac{\mathrm{d}}{\mathrm{d}t} x_t^{\mathsf{T}} = f(t) x_t^{\mathsf{T}} + \frac{g^2(t)}{2\sigma_t} \hat{\epsilon}_c(x_t^{\mathsf{T}} + x_t^{\mathsf{H}}), \tag{6}$$

$$\frac{\mathrm{d}}{\mathrm{d}t} x_t^{\mathsf{H}} = f(t) x_t^{\mathsf{H}} + \frac{g^2(t)}{2\sigma_t} (\omega - 1) \Delta\hat{\epsilon}_c(x_t^{\mathsf{T}} + x_t^{\mathsf{H}}), \tag{7}$$

where $x_T^{\mathsf{T}} = x_T$, $x_T^{\mathsf{H}} = 0$, and $x_t := x_t^{\mathsf{T}} + x_t^{\mathsf{H}}$. The tortoise $x_t^{\mathsf{T}}$ covers the noise estimate part of the diffusion ODE, while the hare $x_t^{\mathsf{H}}$ takes care of the additional guidance term. We call the ODE integrated on the fine-grained grid the tortoise equation (Eq. 6), and the ODE integrated on the coarse grid the hare equation (Eq. 7). Intuitively, the hare equation uses coarser timestep intervals—i.e. larger steps—allowing it to skip unnecessary computation and thus significantly improve the efficiency of integrating the diffusion ODE. Moreover, because both equations retain the standard diffusion ODE form, existing solvers such as DDIM [45] can be applied to each equation without modification.

### 3.2 Tortoise and Hare Guidance

Solving the hare equation (Eq. 7) on the coarse grid is straightforward, since every coarse timestep is also a fine-grained timestep. By contrast, because the tortoise equation (Eq. 6) requires the full state $x_t = x_t^{\mathsf{T}} + x_t^{\mathsf{H}}$ at every fine-grained timestep, we must infer $x_t^{\mathsf{H}}$ at those intermediate points [39]. Instead of using generic extrapolation methods [31], we exploit a property of diffusion model solvers: given $x_t$ and $\hat{\epsilon}_\theta(x_t)$, they can deterministically compute $x_s$ for any $s < t$ by running the chosen solver from $t$ to $s$. From each coarse timestep, we run the solver not only to compute $x_t^{\mathsf{H}}$ for the next coarse timestep but also to populate $x_t^{\mathsf{H}}$ for all intermediate fine-grained timesteps, thereby constructing the full trajectory of $x_t^{\mathsf{H}}$ on the fine-grained grid for use in integrating the tortoise equation.

Building on this formulation, we propose an implementation strategy summarized in Algorithm 1. While the standard diffusion solver evaluates both $\hat{\epsilon}_c(x_t)$ and $\Delta\hat{\epsilon}_c(x_t)$ at every fine-grained timestep, our scheme evaluates $\Delta\hat{\epsilon}_c(x_t)$ only on the coarse grid $C \subset \{t_0, \ldots, t_N\}$, thereby significantly reducing NFE. At each coarse step $t_i \in C$, the updated guidance term is used to integrate the hare

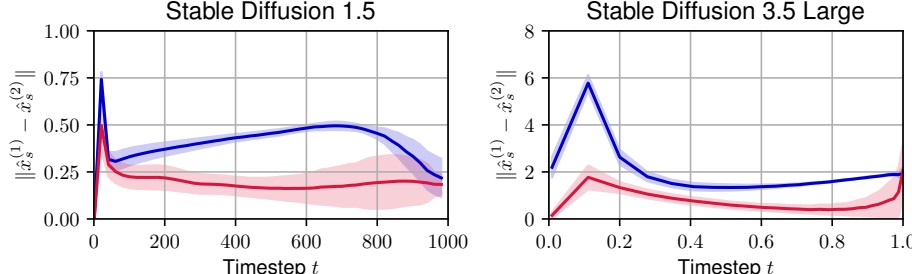

Figure 3: **Approximation error bounds of the tortoise $x_t^\mathsf{T}$ and the hare $x_t^\mathsf{H}$.** We show the per-timestep error bound of the tortoise and the hare terms across sampling steps. The consistently higher bounds for the tortoise curve indicate that the noise estimate is more sensitive to timestep resolution than the additional guidance. Shaded areas denote two standard deviations over multiple prompts.

equation across the fine-grained grid until the next coarse step. We then use the resulting $x_t^\mathsf{H}$ values during the subsequent tortoise equation steps. As a result, the NFE is reduced from $2N$ to $N + |C| - 1$ while preserving the dynamics of the original diffusion ODE. Moreover, it slots seamlessly into existing diffusion pipelines without any changes to their core logic.

### 3.3 Approximation error bound analysis

To determine an appropriate coarse grid $C$ for the hare equation, we now turn to an error-based criterion. Our objective is to ensure that the integration error of $x_t^\mathsf{H}$ remains sufficiently small relative to that of $x_t^\mathsf{T}$. To this end, we adopt a standard multirate strategy [11]. We select coarse step sizes such that the ratio between the hare's approximation error and the tortoise's approximation error does not exceed a user-specified threshold $\rho$ such that $\rho \approx 1$:

$$\frac{\left\|\hat{x}_s^\mathsf{H} - x_s^\mathsf{H}\right\|}{\left\|\hat{x}_s^\mathsf{T} - x_s^\mathsf{T}\right\|} \leq \rho. \tag{8}$$

Here, $x_s^\mathsf{T}$ and $x_s^\mathsf{H}$ denote the analytical solutions to the tortoise and hare equations at timestep $s$, while $\hat{x}_s^\mathsf{T}$ and $\hat{x}_s^\mathsf{H}$ are the corresponding numerical solutions obtained using the diffusion model solver. Given that the solver has order $p$, the local integration error at a single step scales as [14]:

$$\hat{x}_s - x_s = c \cdot (\Delta t)^{p+1} + \mathcal{O}((\Delta t)^{p+2}) \tag{9}$$

where $\Delta t$ is the fine-grained step size and $c$ is an unknown constant. Let the coarse step size be $m\Delta t$, meaning the hare *leaps* $m$ tortoise steps per update. Then, the local integration error of the hare equation over one coarse step becomes:

$$\hat{x}_s^\mathsf{H} - x_s^\mathsf{H} = c^\mathsf{H} \cdot (m\Delta t)^{p+1} + \mathcal{O}\left((\Delta t)^{p+2}\right). \tag{10}$$

In contrast, the tortoise equation accumulates error over $m$ fine-grained steps:

$$\hat{x}_s^\mathsf{T} - x_s^\mathsf{T} = c^\mathsf{T} \cdot m(\Delta t)^{p+1} + \mathcal{O}\left((\Delta t)^{p+2}\right), \tag{11}$$

Taking the ratio from Eq. 8 and ignoring higher-order terms, we obtain:

$$\frac{\left\|\hat{x}_s^\mathsf{H} - x_s^\mathsf{H}\right\|}{\left\|\hat{x}_s^\mathsf{T} - x_s^\mathsf{T}\right\|} = \frac{\left\|c^\mathsf{H}\right\| m^{p+1}(\Delta t)^{p+1}}{\left\|c^\mathsf{T}\right\| m(\Delta t)^{p+1}} = m^p \frac{\left\|c^\mathsf{H}\right\|}{\left\|c^\mathsf{T}\right\|} \leq \rho, \quad \therefore m \leq \left(\rho \left\|c^\mathsf{T}\right\| / \left\|c^\mathsf{H}\right\|\right)^{1/p}. \tag{12}$$

Since $m$ must be a positive integer, we define the maximum allowable value as:

$$m_{\max} := \max\left(1, \left\lfloor \left(\rho \left\|c^\mathsf{T}\right\| / \left\|c^\mathsf{H}\right\|\right)^{1/p} \right\rfloor\right). \tag{13}$$

**Estimating the error constants** To compute $m_{\max}$, we need estimates of $\|c^\mathsf{T}\|$ and $\|c^\mathsf{H}\|$ without relying on the analytic solution $x_s$. We accomplish this using the Richardson extrapolation method [14] . First, solve the ODE once using step size $\Delta t$:

$$\hat{x}_s^{(1)} - x_s = c \cdot (\Delta t)^{p+1} + \mathcal{O}\left((\Delta t)^{p+2}\right). \tag{14}$$

---

**Algorithm 2** Look before you leap

---

**Require:** $m_{\max}(t_i)$            ▷ Calculated $m_{\max}$ for each timestep
**Require:** $\{t_i\}_{0 \leq i \leq N}, t_0 = T, t_N = 0$           ▷ Fine-grained timestep grid
 1: $C \leftarrow \{\}$           ▷ The result is initially an empty set
 2: $i \leftarrow 0$          ▷ Start advancing the fine-grained grid from the first timestep
 3: **while** $i < N$ **do**
 4:      $C \leftarrow C \cup \{t_i\}$           ▷ Add current position
 5:      $i \leftarrow i + m_{\max}(t_i)$           ▷ Advance $m_{\max}(t_i)$ steps
 6: **end while**
 7: $C \leftarrow C \cup \{0\}$           ▷ Include last timestep
 8: **return** $C$

---

Next, solve again using two steps of size $\Delta t/2$:

$$\hat{x}_s^{(2)} - x_s = c \cdot 2(\Delta t/2)^{p+1} + \mathcal{O}\left((\Delta t)^{p+2}\right). \tag{15}$$

Subtracting Eq. 14 and Eq. 15 yields

$$\hat{x}_s^{(1)} - \hat{x}_s^{(2)} = c \cdot \left(1 - 2^{-p}\right)(\Delta t)^{p+1} + \mathcal{O}\left((\Delta t)^{p+2}\right). \tag{16}$$

If we ignore the higher-order terms, the norm of this difference provides a direct estimate proportional to $\|c\|$. We apply this procedure independently to both the tortoise and hare equations to estimate $\|c^{\mathsf{T}}\|$ and $\|c^{\mathsf{H}}\|$, respectively. Empirical results (Fig. 3) on 30,000 prompts from the COCO 2014 dataset [27, 37] show that $\|c^{\mathsf{T}}\|$ is greater than $\|c^{\mathsf{H}}\|$ for most cases, confirming that the tortoise equation is more sensitive to timestep resolution.

**Example usage scenario**    To generate samples using Tortoise and Hare Guidance, we first calculate $m_{\max}$ of each fine-grained timestep (Eq. 13) using the average $\|c^{\mathsf{T}}\|$ and $\|c^{\mathsf{H}}\|$ over a batch of inputs. (More details are covered in Appendix C.) Then we build the coarse timestep grid $C$ via the "look before you leap" strategy (Algorithm 2). Starting at the first fine-grained timestep $t_0$, we insert coarse timesteps so that they lie $m_{\max}(t_i)$ steps ahead, keeping the local error ratio below $\rho$. Finally, we generate samples using Algorithm 1. Note that $C$ could be reused for all subsequent inferences without any additional NFEs.

### 3.4 Adjusting Guidance Scales

Approximating the hare at fine-grained timesteps can lead to a degradation in output quality. To compensate for this, we propose adjusting the guidance scale whenever the additional guidance term is used more than once per timestep. In particular, we introduce a constant boost factor $b$ and scale the guidance term: $\Delta\hat{\epsilon}_c \leftarrow b \cdot \Delta\hat{\epsilon}_c$. This simple multiplicative adjustment improves sample quality, especially in cases where the inner loop (which integrates the hare equation) is repeated multiple times for each coarse step. Our method draws inspiration from prior work such as CFG-Cache [31], which amplifies guidance in the frequency domain using FFT. However, unlike FFT-based methods, our approach avoids the overhead of spectral transforms, which can be computationally expensive for high-dimensional latent variables. The additional guidance term predominantly contains low-frequency information in the early stages of sampling and vice versa [15]. Therefore, selectively enhancing the frequency components of the additional guidance term per timestep has low significance.

Furthermore, CFG and the additional guidance term are of low significance at the later phase of the reverse diffusion process [26, 3]. We leverage this fact by introducing a threshold timestep index $i_{\mathrm{hi}}$ and substituting $\Delta\hat{\epsilon}_c \leftarrow 0$ if $i \geq i_{\mathrm{hi}}$. This simple adjustment helps reduce the NFE even further.

## 4 Experiments

### 4.1 Experimental Settings

**Compared methods**    To demonstrate the effectiveness of our approach, we compare against CFG-Cache [31], a training-free acceleration technique that reuses conditional and unconditional outputs in video diffusion models. Given that CFG-Cache exploits a timestep-adaptive enhancement technique

Table 1: Comparison of methods in terms of distributional similarity and prompt fidelity. Our method is marked in  blue , whereas vanilla CFG is marked in  gray . The best results are **highlighted**.

| Method | NFE ↓ | Distributional similarity | | Prompt fidelity | |
| | | FID ↓ | CMMD ↓ | CS ↑ | IR ↑ |
|---|---|---|---|---|---|
| *Stable Diffusion 1.5 with DDIM* | | | | | |
| CFG [18] | 100 | 14.057 | 0.58885 | 26.294 | 0.14765 |
| CFG-Cache w/o FFT [31] | 70 | 14.240 | **0.59187** | 26.141 | 0.08757 |
| CFG-Cache [31] | 70 | 14.367 | 0.59556 | 26.180 | 0.09735 |
| THG (Ours) | 70 | **14.165** | 0.59223 | **26.189** | **0.11499** |
| *Stable Diffusion 1.5 with DPM-Solver-2 [30]* | | | | | |
| CFG [18] | 100 | 13.255 | 0.60379 | 26.254 | 0.16148 |
| CFG-Cache w/o FFT [31] | 70 | 13.387 | **0.60665** | 26.107 | 0.10513 |
| CFG-Cache [31] | 70 | 13.468 | 0.60880 | 26.147 | 0.11474 |
| THG (Ours) | 70 | **12.909** | 0.60868 | **26.205** | **0.14926** |
| *Stable Diffusion 1.5 with 2nd-order Linear Multistep Method [29]* | | | | | |
| CFG [18] | 100 | 13.540 | 0.60653 | 26.260 | 0.15966 |
| CFG-Cache w/o FFT [31] | 70 | **13.686** | **0.60844** | 26.107 | 0.09881 |
| CFG-Cache [31] | 70 | 13.798 | 0.61142 | 26.144 | 0.10805 |
| THG (Ours) | 70 | **13.686** | 0.61094 | **26.204** | **0.15184** |
| *Stable Diffusion 3.5 Large with Euler method* | | | | | |
| CFG [18] | 56 | 68.158 | 0.81106 | 26.624 | 1.03569 |
| CFG-Cache w/o FFT [31] | 38 | 67.931 | 0.76448 | 26.643 | 1.00715 |
| CFG-Cache [31] | 38 | **67.914** | **0.75324** | 26.668 | 1.00745 |
| THG (Ours) | 38 | 68.252 | 0.80092 | **26.672** | **1.02365** |

| Method | NFE ↓ | Distributional similarity FAD ↓ | Prompt fidelity CLAP Score ↑ |
|---|---|---|---|
| *AudioLDM 2 with DDIM* | | | |
| CFG [18] | 100 | 2.596 | 0.2409 |
| CFG-Cache w/o FFT [31] | 70 | 2.901 | 0.2251 |
| THG (Ours) | 70 | **2.764** | **0.2342** |

to mitigate fine-detail degradation, we evaluate both the full CFG-Cache (with enhancement) and a variant without this enhancement (denoted "CFG-Cache w/o FFT"). All variants are adapted to the diffusion model's modality for a fair comparison. More comparisons with CFG variants are given in Appendix E.

**Implementation details**  We build Tortoise and Hare Guidance with PyTorch [36], Diffusers [48], and Accelerate [13]. We evaluate three pretrained diffusion models—Stable Diffusion 1.5 [40], Stable Diffusion 3.5 Large [47, 10], and AudioLDM 2 [28]. For Stable Diffusion (SD) models, we use prompt–image pairs randomly sampled from COCO 2014 [27, 37]: 30,000 pairs for SD 1.5 and 1,000 pairs for SD 3.5 Large. For AudioLDM 2, we use 2,230 prompt-audio pairs from the validation set of AudioCaps [25]. Experiments are run on a server with an AMD EPYC 74F3 26934-core CPU, 1 TB of RAM, and 8 NVIDIA A100 80GB GPUs. Hyperparameters $(N, \omega, \rho, b, i_{\text{hi}})$ are set to $(50, 7.5, 1.1, 1.1, 38)$ for SD 1.5, $(28, 3.5, 1.0, 1.2, 21)$ for SD 3.5 Large, and $(50, 3.5, 0.9, 1.15, 39)$ for AudioLDM 2.

### 4.2  Main Results

**Quantitative comparison**  Table 1 compares our method to the CFG-Cache variants in terms of distributional similarity metrics such as FID [17, 35], CMMD [22], and FAD [24], together with prompt fidelity metrics such as CLIP Score (CS) [16], ImageReward (IR) [51], and CLAP Score [50] under the same number of function evaluations (NFE). Refer to Appendix F for more details on metrics. On SD 1.5, all methods cut NFE from 100 to 70; ours lowers FID (14.165 vs. 14.240),

*SD 1.5 with DDIM*, Prompt: **A group of zebras grazing in the grass.**

*SD 1.5 with DDIM*, Prompt: **Two cows on a hill above a valley and mountains on the other side.**

*SD 3.5 Large with Euler method*, Prompt: **A single giraffe standing in the middle of tall grass**

*SD 3.5 Large with Euler method*, Prompt: **A bus that sign reads "Crosstown". It is a metro bus.**

|         (a) CFG         |   (b) CFG-Cache w/o FFT   |      (c) CFG-Cache      |      (d) THG (Ours)      |

Figure 4: **Comparison of visual results** for prompts from the COCO 2014 dataset.

matches CMMD, and improves CS and IR over CFG-Cache w/o FFT, and beats full CFG-Cache on CS and IR while keeping FID competitive. On SD 3.5 Large, all cut NFE from 56 to 38; although CFG-Cache slightly leads on FID and CMMD, our method delivers nearly equal FID/CMMD with the highest IR and tied CS. On AudioLDM 2 [28], all cut NFE from 100 to 70; ours lowers FAD (2.764 vs. 2.901) and improves CLAP Score over CFG-Cache w/o FFT. CFG-Cache is excluded since its enhancement is inapplicable to the audio domain. These results show that THG generalizes across solvers and scales, preserving sample distribution and text alignment under aggressive step reduction. The tradeoff of distributional similarity and prompt fidelity is further discussed in Appendix G.

**Qualitative comparison**   Figure 4 compares images generated by our method and the two CFG-Cache variants. The results demonstrate that THG effectively preserves image fidelity and fine details. More visual comparisons are shown in Appendix I.

Table 2: Ablation study for the hyperparameter $b$.

| Method | NFE ↓ | FID ↓ | CMMD ↓ | CS ↑ | IR ↑ |
|---|---|---|---|---|---|
| $b = 1.00$ | 70 | 13.811 | 0.58364 | 26.137 | 0.09395 |
| $b = 1.05$ | 70 | 13.988 | 0.58794 | 26.162 | 0.10456 |
| $b = 1.10$ | 70 | 14.232 | 0.59354 | 26.197 | 0.11576 |
| $b = 1.15$ | 70 | 14.472 | 0.59783 | 26.221 | 0.12639 |
| $b = 1.20$ | 70 | 14.729 | 0.60260 | 26.246 | 0.13478 |

Table 3: Ablation study for the hyperparameter $\rho$.

| Method | NFE ↓ | FID ↓ | CMMD ↓ | CS ↑ | IR ↑ |
|---|---|---|---|---|---|
| $\rho = 0.9$ | 75 | 14.128 | 0.59044 | 26.193 | 0.11942 |
| $\rho = 1.0$ | 73 | 14.148 | 0.59068 | 26.200 | 0.11949 |
| $\rho = 1.1$ | 70 | 14.232 | 0.59354 | 26.197 | 0.11576 |
| $\rho = 1.2$ | 69 | 14.336 | 0.59306 | 26.221 | 0.11262 |
| $\rho = 1.3$ | 67 | 14.280 | 0.59521 | 26.197 | 0.10849 |

## 4.3 Ablation Studies

**Boost factor $b$**  We perform ablation studies on hyperparameters using SD 1.5 with DDIM. Table 2 shows how varying the boost factor $b$ affects inference quality at 70 NFE budget with the same latents $x_T$. As $b$ increases from 1.00 to 1.20, we observe a steady rise in IR from 0.09395 up to 0.13478, indicating stronger image–text alignment, and a modest gain in CS. However, this comes at the cost of higher FID and CMMD values, reflecting a gradual drop in distributional similarity. We select $b = 1.10$ as our default because it strikes the best balance: it substantially boosts IR (0.11576) with only a moderate increase in FID (14.232) and CMMD (0.59354) relative to lower $b$ values.

**Error-ratio threshold $\rho$**  Table 3 summarizes the effect of varying $\rho$ with the same latents $x_T$. Lowering $\rho$ from 1.1 to 0.9 results in more conservative hare leaps—NFE rise from 70 to 75—and yields slightly better FID (14.128 vs. 14.232) and CMMD (0.59044 vs. 0.59354), at the expense of marginally lower IR (0.11942 vs. 0.11576). Increasing $\rho$ to 1.3 reduces NFE to 67 but degrades FID (14.280) and IR (0.10849). We choose $\rho = 1.1$ as our default since it achieves the best trade-off: a 30% NFE reduction (70 NFE) while maintaining competitive fidelity and alignment metrics.

## 5 Conclusion

We present Tortoise and Hare Guidance, a training-free acceleration framework for diffusion sampling that leverages a multirate reformulation of classifier-free guidance (CFG). Exploiting the asymmetric sensitivity of the noise estimate and the additional guidance term to numerical error, Tortoise and Hare Guidance integrates the noise estimate on a fine-grained grid while integrating the additional guidance term on a coarse grid. This approach allows for a substantial reduction in the number of function evaluations (NFE) without sacrificing generation quality. With an error-bound-aware timestep sampler and a guidance scale adjustment, our method achieves up to 30% faster sampling while preserving fidelity across models like Stable Diffusion 1.5, 3.5 Large, and AudioLDM 2, demonstrating the effectiveness of multirate integration for real-time high-quality generation.

**Limitations**  Our experiments are currently limited to latent diffusion models and a few benchmark datasets such as COCO 2014 and AudioCaps. Extending the evaluation to a wider range of architectures, modalities, and downstream tasks will help assess the generality and robustness of our method.

**Broader Impact**  By reducing sampling cost without retraining, Tortoise and Hare Guidance lowers the barrier to deploying diffusion models in real-time applications such as creative tools, accessibility services, and mobile environments. This could result in accelerating the production of synthetic media, including deepfakes and misleading content. Nonetheless, the capabilities of Tortoise and Hare Guidance remain bounded by those of the underlying diffusion model, introducing a limited impact to the quality of such synthetic media.

## Acknowledgments and Disclosure of Funding

We sincerely thank Byeongju Woo, Chanyong Lee, and Eunjin Koh for their constructive discussions and support. We also appreciate Minseo Kim and Minkyu Song for providing insightful feedback. This work was supported by the Agency For Defense Development Grant Funded by the Korean Government (912A45701).

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

# A  $v$-prediction models

Recent models such as Stable Diffusion 3.5 [47] directly infer $v$, or the *velocity field* of the reverse diffusion process. The diffusion ODE is then defined as

$$\frac{\mathrm{d}}{\mathrm{d}t}x_t = \hat{v}_\theta(x_t), \quad x_T \sim \mathcal{N}(0, I). \tag{17}$$

By the definition of CFG [18], we have

$$\hat{v}_\theta(x_t) := \hat{v}_\varnothing(x_t) + \omega \cdot (\hat{v}_c(x_t) - \hat{v}_\varnothing(x_t)) \equiv \hat{v}_c(x_t) + (\omega - 1) \cdot \Delta\hat{v}_c(x_t) \tag{18}$$

where $\Delta\hat{v}_c(x_t) := \hat{v}_c(x_t) - \hat{v}_\varnothing(x_t)$. Substituting Eq. 18 into Eq. 17 yields the following:

$$\frac{\mathrm{d}}{\mathrm{d}t}x_t = \hat{v}_c(x_t) + (\omega - 1) \cdot \Delta\hat{v}_c(x_t). \tag{19}$$

We split this diffusion ODE into a multirate system of ODEs similar to Section 3.1.

$$\frac{\mathrm{d}}{\mathrm{d}t}x_t^{\mathsf{T}} = \hat{v}_c(x_t^{\mathsf{T}} + x_t^{\mathsf{H}}), \quad \frac{\mathrm{d}}{\mathrm{d}t}x_t^{\mathsf{H}} = (\omega - 1) \cdot \Delta\hat{v}_c(x_t^{\mathsf{T}} + x_t^{\mathsf{H}}). \tag{20}$$

Both equations retain the form of Eq. 17 so that existing solvers as the Euler method can be applied to each equation without modification. Furthermore, Algorithm 1 could be utilized unchanged since it is agnostic to the form of equation or the type of the diffusion model solver.

# B  Proof for approximation error bound analysis

We provide a proof for error accumulation presented in Section 3.3. More rigourous analysis of error bounds could be found in Section II. 3. of [14].

**Theorem 1.** *Assume the local integration error of an ODE using a solver of order $p$ and timestep size $\Delta t$ is given by:*

$$\hat{x}_{t-\Delta t} - x_{t-\Delta t} = c \cdot (\Delta t)^{p+1} + \mathcal{O}((\Delta t)^{p+2}) \tag{21}$$

*for sufficiently small $\Delta t$. Then the error of using the same solver repeatedly for $m$ steps is given by*

$$\hat{x}_{t-m\Delta t} - x_{t-m\Delta t} = c \cdot m(\Delta t)^{p+1} + \mathcal{O}((\Delta t)^{p+2}). \tag{22}$$

*Proof.* We use mathematical induction. (**Base step**) For $m = 1$, Eq. 22 reduces to the assumption. (**Inductive step**) Assume the error of using the solver $m$ times is given by Eq. 22. We proceed to the next iteration to obtain $\hat{x}_{t-(m+1)\Delta t}$. Let $\tilde{x}_{t-(m+1)\Delta t}$ be the *exact* solution given by solving the ODE from $t - m\Delta t$ to $t - (m+1)\Delta t$ using $\hat{x}_{t-m\Delta t}$. The error in Eq. 22 is transported to the next timestep as

$$\tilde{x}_{t-(m+1)\Delta t} - x_{t-(m+1)\Delta t} = (I + \mathcal{O}(\Delta t))\,(\hat{x}_{t-m\Delta t} - x_{t-m\Delta t}) \tag{23}$$

$$= c \cdot m(\Delta t)^{p+1} + \mathcal{O}((\Delta t)^{p+2}). \tag{24}$$

On the other hand, the local error of the next iteration is also given by Eq. 21:

$$\hat{x}_{t-(m+1)\Delta t} - \tilde{x}_{t-(m+1)\Delta t} = c \cdot (\Delta t)^{p+1} + \mathcal{O}((\Delta t)^{p+2}). \tag{25}$$

The error of using the solver $m + 1$ times is thus

$$\hat{x}_{t-(m+1)\Delta t} - x_{t-(m+1)\Delta t} = c \cdot (m+1)(\Delta t)^{p+1} + \mathcal{O}((\Delta t)^{p+2}). \tag{26}$$

Therefore the error of using the ODE solver $m$ times is given by Eq. 22 for all positive integer $m$.  □

# C  More details for Richardson Extrapolation

We specify further details about the computation of the coarse timestep grid $C$. We calculate $\|\hat{x}_s^{\mathsf{T}(1)} - \hat{x}_s^{\mathsf{T}(2)}\|$ and $\|\hat{x}_s^{\mathsf{H}(1)} - \hat{x}_s^{\mathsf{H}(2)}\|$ by solving both the tortoise and hare equations on the fine-grained timestep grid using Algorithm 3. In particular, for each denoising step $t_i$, we first find $\hat{x}_{t_{i+1}}^{(1)}$ by using the diffusion model solver once from $t_i$ to $t_{i+1}$. Then we find $\hat{x}_{t_{i+1}}^{(2)}$ by using the diffusion model solver twice, from $t_i$ to $(t_i + t_{i+1})/2$ and from $(t_i + t_{i+1})/2$ to $t_{i+1}$. We use $\hat{x}_{t_{i+1}}^{(1)}$ for the next denoising step to ensure that we follow the reference trajectory of CFG [18]. Together with Algorithm 2, we obtain the coarse timestep grid $C$ specified in Table 4.

**Algorithm 3** Richardson Extrapolation

**Require:** $x_T \sim \mathcal{N}(0, \sigma_T^2 I)$  ▷ Initial noise
**Require:** $\omega \geq 0$  ▷ Guidance scale
**Require:** $\{t_i\}_{0 \leq i \leq N}, t_0 = T, t_N = 0$  ▷ Fine-grained timestep grid
1: $x_T^{\mathsf{T}} \leftarrow x_T$
2: $x_T^{\mathsf{H}} \leftarrow 0$
3: **for** $i = 0$ **to** $N - 1$ **do**
4: $\quad \hat{\epsilon}_c \leftarrow \hat{\epsilon}_\theta(x_{t_i}^{\mathsf{T}} + x_{t_i}^{\mathsf{H}}, c)$
5: $\quad \hat{\epsilon}_\varnothing \leftarrow \hat{\epsilon}_\theta(x_{t_i}^{\mathsf{T}} + x_{t_i}^{\mathsf{H}}, \varnothing)$
6: $\quad \Delta\hat{\epsilon}_c \leftarrow \hat{\epsilon}_c - \hat{\epsilon}_\varnothing$
7: $\quad \hat{x}_{t_{i+1}}^{\mathsf{T}(1)} \leftarrow \text{Solver}(x_{t_i}^{\mathsf{T}}, \hat{\epsilon}_c, t_i, t_{i+1})$  ▷ $\hat{x}_{t_{i+1}}^{(1)}$ of the tortoise
8: $\quad \hat{x}_{t_{i+1}}^{\mathsf{H}(1)} \leftarrow \text{Solver}(x_{t_i}^{\mathsf{H}}, (\omega - 1) \cdot \Delta\hat{\epsilon}_c, t_i, t_{i+1})$  ▷ $\hat{x}_{t_{i+1}}^{(1)}$ of the hare
9: $\quad t_m = (t_i + t_{i+1})/2$  ▷ Midpoint of current and next timesteps
10: $\quad \hat{x}_{t_m}^{\mathsf{T}(2)} \leftarrow \text{Solver}(x_{t_i}^{\mathsf{T}}, \hat{\epsilon}_c, t_i, t_m)$
11: $\quad \hat{x}_{t_m}^{\mathsf{H}(2)} \leftarrow \text{Solver}(x_{t_i}^{\mathsf{H}}, (\omega - 1) \cdot \Delta\hat{\epsilon}_c, t_i, t_m)$
12: $\quad \hat{\epsilon}_c \leftarrow \hat{\epsilon}_\theta \left( \hat{x}_{t_m}^{\mathsf{T}(2)} + \hat{x}_{t_m}^{\mathsf{H}(2)}, c \right)$
13: $\quad \hat{\epsilon}_\varnothing \leftarrow \hat{\epsilon}_\theta \left( \hat{x}_{t_m}^{\mathsf{T}(2)} + \hat{x}_{t_m}^{\mathsf{H}(2)}, \varnothing \right)$
14: $\quad \Delta\hat{\epsilon}_c \leftarrow \hat{\epsilon}_c - \hat{\epsilon}_\varnothing$
15: $\quad \hat{x}_{t_{i+1}}^{\mathsf{T}(2)} \leftarrow \text{Solver}(\hat{x}_{t_m}^{\mathsf{T}(2)}, \hat{\epsilon}_c, t_m, t_{i+1})$  ▷ $\hat{x}_{t_{i+1}}^{(2)}$ of the tortoise
16: $\quad \hat{x}_{t_{i+1}}^{\mathsf{H}(2)} \leftarrow \text{Solver}(\hat{x}_{t_m}^{\mathsf{H}(2)}, (\omega - 1) \cdot \Delta\hat{\epsilon}_c, t_m, t_{i+1})$  ▷ $\hat{x}_{t_{i+1}}^{(2)}$ of the hare
17: $\quad x_{t_{i+1}}^{\mathsf{T}} \leftarrow \hat{x}_{t_{i+1}}^{\mathsf{T}(1)}$  ▷ Tortoise of next step
18: $\quad x_{t_{i+1}}^{\mathsf{H}} \leftarrow \hat{x}_{t_{i+1}}^{\mathsf{H}(1)}$  ▷ Hare of next step
19: **end for**
20: **return** $\|\hat{x}_{t_{i+1}}^{\mathsf{T}(1)} - \hat{x}_{t_{i+1}}^{\mathsf{T}(2)}\|, \|\hat{x}_{t_{i+1}}^{\mathsf{H}(1)} - \hat{x}_{t_{i+1}}^{\mathsf{H}(2)}\|$

Table 4: Obtained coarse timestep grid for different $\rho$ values. Our choice is marked in blue . For brevity, only indices of the timesteps are shown. Note that only $i < i_{\text{hi}}$ is actually used in the final algorithm.

| $\rho$ | $\{i \mid t_i \in C\}$ |
|---|---|
| *Stable Diffusion 1.5 with DDIM* | |
| 0.9 | {0, 1, 2, 3, 4, 5, 6, 7, 8, 10, 12, 14, 16, 18, 20, 22, 24, 26, 28, 30, 32, 34, 35, 36, 37, 38, 39, 40, 41, 42, 43, 44, 45, 46, 47, 48, 49} |
| 1.0 | {0, 1, 2, 3, 4, 5, 6, 7, 9, 11, 13, 15, 17, 19, 21, 23, 25, 27, 29, 31, 33, 35, 37, 38, 39, 40, 41, 42, 43, 44, 45, 46, 47, 48, 49} |
| 1.1 | {0, 1, 2, 3, 4, 5, 6, 8, 10, 12, 14, 17, 20, 23, 26, 28, 30, 32, 34, 36, 38, 39, 40, 41, 42, 43, 44, 45, 46, 47, 48, 49} |
| 1.2 | {0, 1, 2, 3, 4, 5, 7, 9, 11, 14, 17, 20, 23, 26, 29, 31, 33, 35, 37, 39, 41, 42, 43, 44, 45, 46, 47, 48, 49} |
| 1.3 | {0, 1, 2, 3, 4, 6, 8, 10, 13, 16, 19, 22, 25, 28, 31, 34, 36, 38, 40, 42, 44, 45, 46, 47, 48, 49} |
| *Stable Diffusion 3.5 Large with Euler method* | |
| 0.9 | {0, 1, 2, 3, 5, 7, 10, 13, 16, 18, 20, 22, 23, 24, 25, 26} |
| 1.0 | {0, 1, 2, 4, 6, 9, 12, 15, 18, 20, 22, 23, 24, 25, 26} |
| 1.1 | {0, 1, 2, 4, 6, 9, 13, 17, 20, 22, 23, 24, 25, 27} |
| 1.2 | {0, 1, 2, 4, 7, 11, 15, 19, 21, 23, 25, 27} |
| 1.3 | {0, 1, 3, 6, 10, 15, 19, 22, 24, 26} |

Table 5: Obtained coarse timestep grid for different $\omega$ values. Our choice is marked in `blue`. For brevity, only indices of the timesteps are shown. Note that only $i < i_{\text{hi}}$ is actually used in the final algorithm. The results show that while a bigger $\omega$ results in a denser $C$, the overall trend is consistent.

| Variant | $\{i\|t_i \in C\}$ |
|---|---|
| *Stable Diffusion 1.5 with DDIM* | |
| $\omega = 6.5, \rho = 0.93$ | {0, 1, 2, 3, 4, 5, 6, 8, 10, 12, 14, 17, 20, 23, 26, 28, 30, 32, 34, 36, 38, 39, 40, 41, 42, 43, 44, 45, 46, 47, 48, 49, 50} |
| $\omega = 6.5, \rho = 1.1$ | {0, 1, 2, 3, 4, 6, 8, 10, 13, 16, 19, 22, 25, 28, 31, 33, 35, 37, 39, 41, 43, 44, 45, 46, 47, 48, 49, 50} |
| $\omega = 7.5, \rho = 1.1$ | {0, 1, 2, 3, 4, 5, 6, 8, 10, 12, 14, 17, 20, 23, 26, 28, 30, 32, 34, 36, 38, 39, 40, 41, 42, 43, 44, 45, 46, 47, 48, 49, 50} |
| $\omega = 8.5, \rho = 1.1$ | {0, 1, 2, 3, 4, 5, 6, 7, 8, 10, 12, 14, 16, 18, 20, 22, 24, 26, 28, 30, 32, 34, 36, 37, 38, 39, 40, 41, 42, 43, 44, 45, 46, 47, 48, 49, 50} |
| $\omega = 8.5, \rho = 1.22$ | {0, 1, 2, 3, 4, 5, 6, 8, 10, 12, 14, 17, 20, 23, 26, 28, 30, 32, 34, 36, 38, 39, 40, 41, 42, 43, 44, 45, 46, 47, 48, 49, 50} |

Table 6: Obtained coarse timestep grid with fewer sample trajectories. Our original choice is marked in `blue`. For brevity, only indices of the timesteps are shown. Note that only $i < i_{\text{hi}}$ is actually used in the final algorithm. The results show that $C$ can be computed accurately with only 1,000 sample trajectories.

| Batch size | IoU | $\{i\|t_i \in C\}$ |
|---|---|---|
| *Stable Diffusion 1.5 with DDIM* | | |
| 30k | – | {0, 1, 2, 3, 4, 5, 6, 8, 10, 12, 14, 17, 20, 23, 26, 28, 30, 32, 34, 36, 38, 39, 40, 41, 42, 43, 44, 45, 46, 47, 48, 49, 50} |
| 1k (trial 1) | 100% | {0, 1, 2, 3, 4, 5, 6, 8, 10, 12, 14, 17, 20, 23, 26, 28, 30, 32, 34, 36, 38, 39, 40, 41, 42, 43, 44, 45, 46, 47, 48, 49, 50} |
| 1k (trial 2) | 100% | {0, 1, 2, 3, 4, 5, 6, 8, 10, 12, 14, 17, 20, 23, 26, 28, 30, 32, 34, 36, 38, 39, 40, 41, 42, 43, 44, 45, 46, 47, 48, 49, 50} |
| 1k (trial 3) | 100% | {0, 1, 2, 3, 4, 5, 6, 8, 10, 12, 14, 17, 20, 23, 26, 28, 30, 32, 34, 36, 38, 39, 40, 41, 42, 43, 44, 45, 46, 47, 48, 49, 50} |
| | . . . | |
| 1k (trial 8) | 57.5% | {0, 1, 2, 3, 4, 5, 7, 9, 11, 13, 16, 19, 22, 25, 28, 30, 32, 34, 36, 38, 39, 40, 41, 42, 43, 44, 45, 46, 47, 48, 49, 50} |
| | . . . | |
| 1k (trial 30) | 100% | {0, 1, 2, 3, 4, 5, 6, 8, 10, 12, 14, 17, 20, 23, 26, 28, 30, 32, 34, 36, 38, 39, 40, 41, 42, 43, 44, 45, 46, 47, 48, 49, 50} |

# D   Sensitivity of $C$

**Guidance weight** $\omega$   The coarse timestep grid $C$ depends on guidance weight $\omega$, since the error bounds computed by Algorithm 3 depend on $\omega$. As $\omega$ increases, both the hare term and its approximation error grow, resulting in a denser coarse grid $C$. Table 5 shows $C$ evaluated under different $\omega$ values. While a bigger guidance scale results in a denser $C$, the overall trend is consistent; one can obtain the same $C$ by adjusting $\rho$.

**Batch size**   While we computed $C$ with sample trajectories on 30,000 prompts from the COCO 2014 dataset, it is possible to compute $C$ using fewer sample trajectories. Table 6 shows $C$ computed on a batch of 1,000 prompts, compared to $C$ computed on 30,000 prompts. The results closely matched our original, large-scale estimate. Compared to the original estimate, the 1,000 sample estimate demonstrated 95.25% IoU (i.e., Jaccard index) in average over 30 trials.

Table 7: Comparison of methods in terms of distributional similarity and prompt fidelity. Our method is marked in blue, whereas vanilla CFG is marked in gray. The parameters of THG correspond to $(\rho, b, i_{\text{hi}})$. The best results are **highlighted**.

| Method | $N$ | NFE $\downarrow$ | Distributional similarity | | Prompt fidelity | |
| --- | --- | --- | --- | --- | --- | --- |
| | | | FID $\downarrow$ | CMMD $\downarrow$ | CS $\uparrow$ | IR $\uparrow$ |
| *Stable Diffusion 1.5 with DDIM* | | | | | | |
| CFG [18] | 50 | 100 | 14.133 | 0.58948 | 26.295 | 0.14764 |
| Selective Guidance [12] | 50 | 70 | **12.895** | **0.56052** | 25.602 | −0.06691 |
| Guidance Interval [26] | 50 | 70 | 14.555 | 0.62227 | 26.153 | 0.09658 |
| CFG-Cache [31] | 50 | 70 | 14.422 | 0.59759 | 26.179 | 0.09705 |
| THG (1.1, 1.1, 38) | 50 | 70 | 14.232 | 0.59354 | **26.197** | **0.11576** |
| CFG [18] | 35 | 70 | 13.342 | 0.57232 | 26.321 | 0.12694 |
| Selective Guidance [12] | 35 | 49 | **12.299** | **0.54550** | 25.623 | −0.09007 |
| Guidance Interval [26] | 35 | 49 | 14.490 | 0.61178 | 26.140 | 0.05919 |
| CFG-Cache [31] | 35 | 49 | 13.773 | 0.58162 | 26.120 | 0.05246 |
| THG (1.7, 1.1, 30) | 35 | 49 | 13.468 | 0.57637 | **26.265** | **0.09202** |
| CFG [18] | 20 | 40 | 13.366 | 0.56159 | 26.370 | 0.10090 |
| Selective Guidance [12] | 20 | 30 | **12.839** | **0.54740** | 25.900 | −0.04452 |
| Guidance Interval [26] | 20 | 30 | 14.937 | 0.61333 | 26.244 | 0.02827 |
| CFG-Cache [31] | 20 | 30 | 13.675 | 0.56815 | 26.211 | 0.04604 |
| THG (3.5, 1.1, 17) | 20 | 30 | 13.345 | 0.56719 | **26.278** | **0.07212** |

# E  More comparisons with CFG variants

Table 7 extends our comparison to include additonal baselines [12, 26] across a wider range of NFEs. Note that we used a different number of fine-grained timesteps $N$ to control the NFE of CFG. The results demonstrate that our approach achieves superior image quality with the same NFE budget. While Selective Guidance [12] shows lower FID and CMMD values, they generate images with low prompt fidelity and degraded details indicated by lower CS and IR values as noted in Appendix G. Aside from this phenomenon, THG achieves superior overall performance.

**Comparison with vanilla CFG**  Although vanilla CFG with 70 NFE yields a FID of 13.342 (better than THG's 14.232 with 70 NFE), its PSNR drops substantially to 19.42 dB compared to 24.16 dB for THG. This aligns with observations in [6], where reducing $N$ sometimes lowers FID. Moreover, [35, 23] show that FID may not reliably reflect perceptual quality when the image structures diverge, so we focus on comparisons at the same $N$.

# F  More details for metrics

**FID, FAD**  Fréchet Inception Distance [17] (FID) is a ubiquitously used metric for developing and adopting image generative models. It measures the distance between real and generated images in a deep feature space to capture relevant features of the two distributions [35]. Therefore, a lower FID value indicates realistic generation. To measure FID, one first uses an InceptionV3 feature extractor model to compute features from real and generated images. Under the assumption that the resulting feature sets follows a multidimensional Gaussian distribution, the distance of the two distributions $\mathcal{N}(\mu_r, \Sigma_r)$ and $\mathcal{N}(\mu_g, \Sigma_g)$ is given by the Fréchet distance

$$\text{FD} = ||\mu_r - \mu_g||_2^2 + \text{tr}(\Sigma_r + \Sigma_g - 2(\Sigma_r \Sigma_g)^{1/2}). \tag{27}$$

Similarly, Fréchet Audio Distance [24] (FAD) measures the distance between real and generated audio by using a VGGish embedding model to extract features from audio clips.

**CMMD**  CMMD [22] is a recently proposed alternative for FID. It uses CLIP [38] embeddings and the Maximum Mean Discrepancy (MMD) distance instead of InceptionV3 features and the Fréchet distance. CLIP is trained on 400 million images with corresponding text descriptions and therefore

Table 8: Ablation study for the guidance scale $\omega$ with CFG [18].

| Method | NFE ↓ | FID ↓ | CMMD ↓ | CS ↑ | IR ↑ |
|---|---|---|---|---|---|
| *Stable Diffusion 1.5 with DDIM* | | | | | |
| $\omega = 2.5$ | 100 | 8.438 | 0.56672 | 25.153 | -0.28577 |
| $\omega = 3.5$ | 100 | 9.143 | 0.54192 | 25.687 | -0.09190 |
| $\omega = 4.5$ | 100 | 10.644 | 0.54764 | 25.935 | 0.00670 |
| $\omega = 5.5$ | 100 | 12.030 | 0.56171 | 26.110 | 0.07195 |
| $\omega = 6.5$ | 100 | 13.222 | 0.57673 | 26.225 | 0.11582 |
| $\omega = 7.5$ | 100 | 14.133 | 0.58948 | 26.295 | 0.14764 |
| $\omega = 8.5$ | 100 | 14.902 | 0.60343 | 26.369 | 0.17431 |

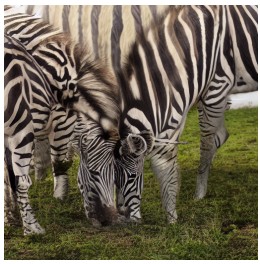 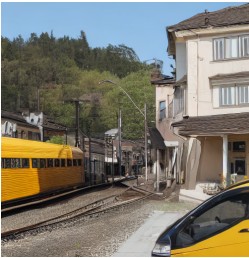 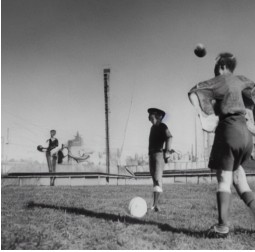 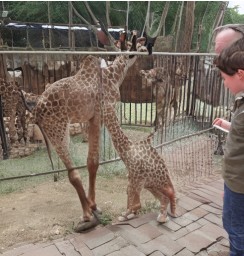

Figure 5: **Generated images using** $\omega = 2.5$ for the prompts "A group of zebras grazing in the grass.", "A yellow commuter train traveling past some houses.", "A couple of men standing on a field playing baseball.", and "Zoo scene of children at zoo near giraffes, attempting to pet or feed them." from the COCO 2014 dataset.

much more suitable for capturing rich and diverse content. MMD does not make any assumptions about the underlying distributions, unlike the Fréchet distance which assumes multidimensional Gaussian distributions. By combining CLIP and MMD, CMMD avoids the drawbacks of FID.

**CLIP Score, CLAP Score** CLIP Score [16] uses CLIP [38] to assess image-caption compatibility. CLIP learns a multimodal embedding space by jointly traning an image encoder and a text encoder, while the cosine similarity of matching image and text embeddings are maximized. Leveraging this design, CLIP score is defined by the cosine similarity of the image embedding $E_I$ and text prompt embedding $E_C$ as

$$\text{CLIPScore}(I, C) = \max(100 \cdot \cos(E_I, E_C), 0). \tag{28}$$

We report the average CLIP score over the generated image set. Similarly, CLAP Score [50] uses the CLAP model to assess audio-caption compatibility.

**ImageReward** ImageReward [51] is a general-purpose text-to-image human preference reward model, effectively encoding human preferences by traning on actual human feedback. Experiments shows that ImageReward aligns with human ranking better than zero-shot FID and CLIP Score. Also, ImageReward values have larger quantile value range than that of CLIP Score, demonstrating that it can well distinguish the quality of images from each other. We report the average ImageReward value over the generated image set.

## G Tradeoff of distributional similarity and prompt fidelity

Tables 1 and 2 demonstrate a tradeoff between distributional similarity metrics (FID, CMMD) and prompt fidelity metrics (CS, IR). When the prompt fidelity metrics improve so that each image matches better with the given prompt, the distributional similarity metrics worsen so that the distribution of the images is further from that of real images.

We further investigate this phenomenon by conducting an additional ablation study for the guidance scale $\omega$ using Stable Diffusion 1.5 and CFG. Table 8 shows how the metrics change as $\omega$ is changed. The minimum FID is achieved at $\omega = 2.5$ and the minimum CMMD is achieved at $\omega = 3.5$. However,

they suffer from low CS and IR. Generated images using $\omega = 2.5$ are visualized in Fig. 5, showing degraded details or insufficient text alignment. This suggests that lower FID or CMMD does not always indicate better generation quality. While these distributional similarity metrics measure both image plausibility and diversity, they can possibly fail to report high-quality details of the images with lower values.

Since the global structure of each image is determined by the initial few steps of the reverse diffusion process [26, 3], the images generated by the methods in Table 1 have mostly shared global structures and differ on delicate details. Given that, we suggest that the human-perceived quality of generated samples could be better explained by the prompt fidelity metrics compared to the distributional similarity metrics. Our results in Table 1 with slightly higher FID or CMMD therefore do not indicate a significant degradation of generation quality.

## H  Licenses

- **Stable Diffusion 1.5** – weights released under the CreativeML Open RAIL-M license (v1.0; `https://github.com/CompVis/stable-diffusion/blob/main/LICENSE`)

- **Stable Diffusion 3.5 Large** – weights released under the Stability AI Community Licence v3 (research & commercial use for organizations or individuals with < USD 1 M annual revenue; `https://stability.ai/license`)

- **FID** – clean-FID implementation by Parmar et al., released under the MIT License (v1.0; `https://github.com/GaParmar/clean-fid/blob/main/LICENSE`)

- **CMMD** – PyTorch implementation of CLIP Maximum Mean Discrepancy by Sayak Paul, released under the Apache License 2.0 (v2.0; `https://github.com/sayakpaul/cmmd-pytorch/blob/main/LICENSE`)

- **CLIP Score** – TorchMetrics' CLIPScore module released under the Apache License 2.0 (v2.0; `https://github.com/Lightning-AI/metrics/blob/master/LICENSE`; Lightning-AI)

- **ImageReward** – model and evaluation code released under the Apache License 2.0 (v2.0; `https://github.com/THUDM/ImageReward/blob/main/LICENSE`; Xu et al., 2023)

- **MS COCO 2014**:
  - Annotations released under the Creative Commons Attribution 4.0 International license (CC BY 4.0; `https://creativecommons.org/licenses/by/4.0/`)
  - Underlying images governed by Flickr Terms of Use; users must comply with Flickr's rules when reusing or redistributing any COCO images.

- **AudioLDM 2** – weights released under the Creative Commons Attribution–NonCommercial–ShareAlike 4.0 International License (`https://github.com/haoheliu/AudioLDM2/blob/main/LICENSE`)

- **AudioCaps** – dataset released under the MIT License (v1.0; `https://github.com/cdjkim/audiocaps/blob/master/LICENSE`)

- **FAD** – PyTorch implementation of Frechet Audio Distance by Hao Hao Tan, released under the MIT License (v1.0; `https://github.com/gudgud96/frechet-audio-distance/blob/main/LICENSE`)

- **CLAP Score** – model and evaluation code released under the Creative Commons CC0 1.0 Universal License; public domain dedication (`https://github.com/LAION-AI/CLAP/blob/main/LICENSE`)

## I  More qualitative results

Figure 6 shows more qualitative results for Stable Diffusion 1.5. Figures 7 and 8 show more qualitative results for Stable Diffusion 3.5 Large.

Prompt: **Two horses are frolicking as spectators take pictures.**

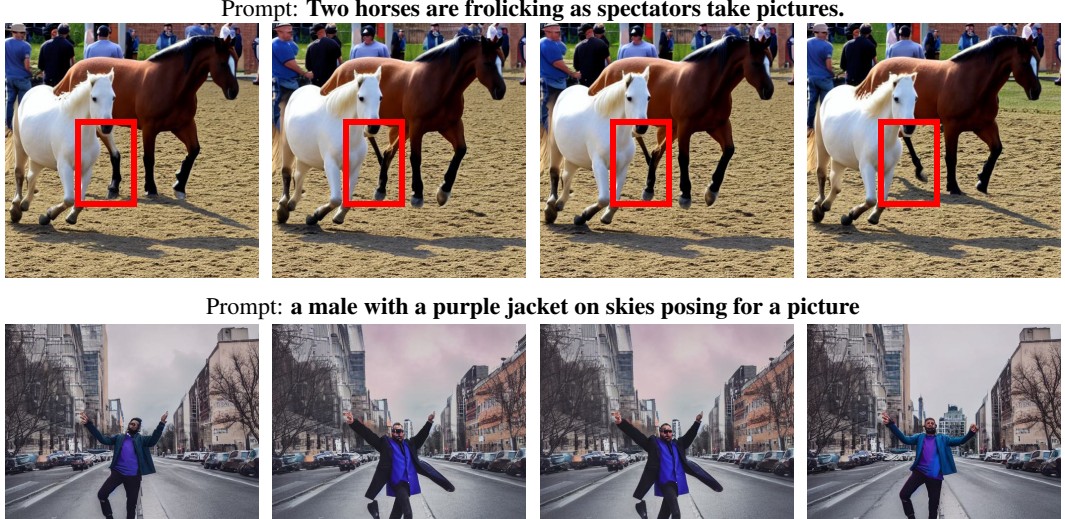

Prompt: **a male with a purple jacket on skies posing for a picture**

|      (a) CFG (baseline)      |   (b) CFG-Cache w/o FFT   |   (c) CFG-Cache   |   (d) Ours   |
|    NFE = 100    |    NFE = 70    |    NFE = 70    |    NFE = 70    |

Figure 6: **Comparison of visual results** for prompts from the COCO 2014 dataset using Stable Diffusion 1.5.

Prompt: **A woman and a man are playing the nintendo wii video game system**

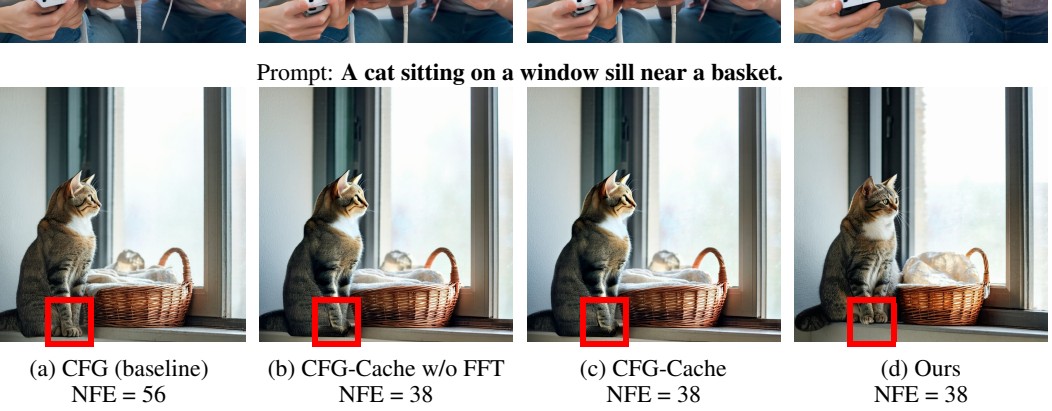

Prompt: **A cat sitting on a window sill near a basket.**

|      (a) CFG (baseline)      |   (b) CFG-Cache w/o FFT   |   (c) CFG-Cache   |   (d) Ours   |
|    NFE = 56    |    NFE = 38    |    NFE = 38    |    NFE = 38    |

Figure 7: **Comparison of visual results** for prompts from the COCO 2014 dataset using Stable Diffusion 3.5 Large.

Prompt: **A red fire hydrant is set up in a grassy clearing.**

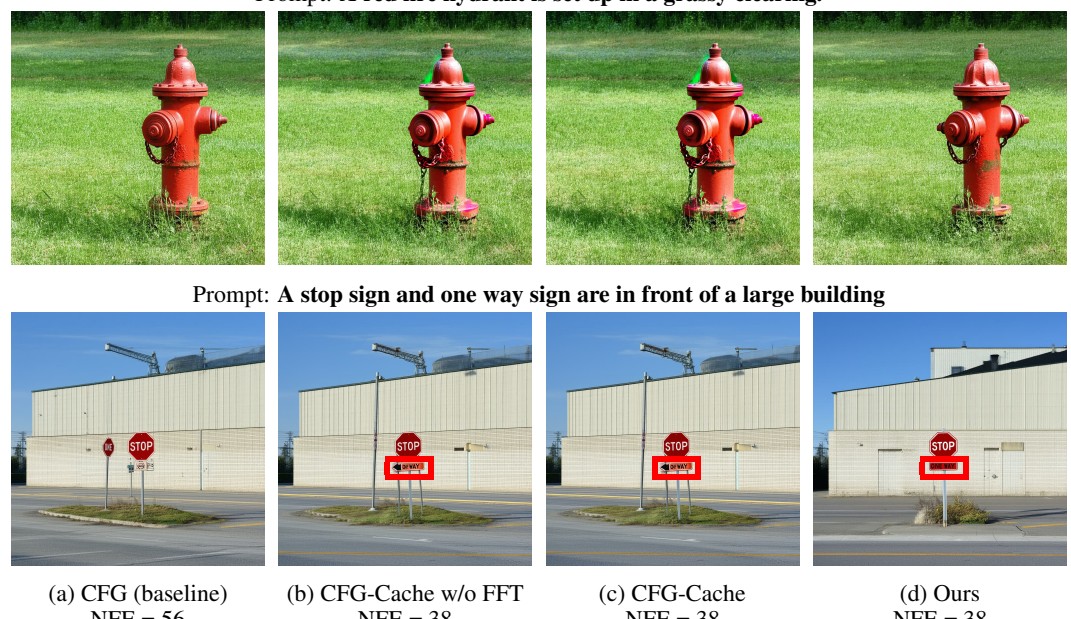

Prompt: **A stop sign and one way sign are in front of a large building**

(a) CFG (baseline)          (b) CFG-Cache w/o FFT          (c) CFG-Cache          (d) Ours
NFE = 56                    NFE = 38                       NFE = 38               NFE = 38

Figure 8: **Comparison of visual results** for prompts from the COCO 2014 dataset using Stable Diffusion 3.5 Large.

