# OpenReview forum: "Tortoise and Hare Guidance: Accelerating Diffusion Model Inference with Multirate Integration"
_NeurIPS.cc/2025/Conference — NeurIPS 2025 poster_

### Official Review · Reviewer_FF6M · 2025-07-02

**Clarity:** 3
**Significance:** 2
**Originality:** 3
**Rating:** 5
**Confidence:** 4

**Summary:**

This paper introduces Tortoise and Hare Guidance (THG), a training-free sampling acceleration method for diffusion model. THG reformulates classifier-free guidance (CFG) by utilizing the asymmetric sensitivity of the noise estimate (tortoise branch) and the additional guidance term (hare branch) to numerical error, reducing the number of function evaluations (NFE) by up to 30% with small loss in sample quality. THG achieves good performance on Stable Diffusion 1.5 and 3.5 Large.

**Questions:**

- Though choosing the coarse time schedule based on error bound analysis is convincing, it takes considerable efforts to estimate the error constants. How is the performance gain of this carefully designed schedule? Could a naive choice also provide good performance? A naive choice of the coarse time schedule can also make the use of higher-order solvers reasonable.
- It could be better to include the latency comparison with existing method to show the superiority of the proposed method.

**Ethical Concerns:**

["NO or VERY MINOR ethics concerns only"]

**Final Justification:**

Most of my concerns are addressed by these new quantative results. I decide to increase my rating.

**Limitations:**

Yes.

**Paper Formatting Concerns:**

I don't find any major formatting issues in this paper.

**Quality:**

3

**Strengths And Weaknesses:**

Strengths:
- The paper is well-organized and the story is interesting. The proposed method is built upon multirate formulation with error bound analyzed for choosing proper coarse time schedule, which is novel.
- THG achieves considerable acceleration without training, making it applicable to existing diffusion pipelines.
- Experiments on Stable Diffusion models demonstrate competitive performance in terms of FID, CLIP Score, and ImageReward compared to the baseline.

Weaknesses:
- This paper only includes CFG-cache for comparison. However, there are a series of works [1, 2, 3], which are also mentioned in Section 2, focusing on accelerating CFG sampling but are not quantitatively compared.
- Gradually increasing the guidance scale by $b$ is kind of wired, as it breaks the fair comparison with baseline where the guidance scale is fixed.
- The experiment could be conducted more thoroughly. Currently, each model only has one choice of NFE. Including more choices like 20 or 10 steps could further enrich the results and demonstrate the superiority of THG.

References

[1] Angela Castillo, Jonas Kohler, Juan C Pérez, Juan Pablo Pérez, Albert Pumarola, Bernard Ghanem, Pablo Arbeláez, and Ali Thabet. Adaptive guidance: Training-free acceleration of conditional diffusion models. arXiv preprint arXiv:2312.12487, 2023.

[2] Pareesa Ameneh Golnari, Zhewei Yao, and Yuxiong He. Selective guidance: Are all the denoising steps of guided diffusion important? arXiv preprint arXiv:2305.09847, 2023.

[3] T. Kynkäänniemi, M. Aittala, T. Karras, S. Laine, T. Aila, and J. Lehtinen. Applying guidance in a limited interval improves sample and distribution quality in diffusion models. In Proc. NeurIPS, 2024.

---

> ### Author Rebuttal · Authors · 2025-07-31
>
> Dear reviewer FF6M,
>
> Thank you for your valuable feedback and constructive comments. We appreciate your attention to the novelty and applicability of THG. We hereby address your concerns and questions.
>
> > * This paper only includes CFG-cache for comparison. However, there are a series of works [1, 2, 3], which are also mentioned in Section 2, focusing on accelerating CFG sampling but are not quantitatively compared.
>
> [A1] We sincerely thank the reviewer for this suggestion. In response, we extended our experimental comparison to include the acceleration techniques proposed in [2, 3]. This demonstrates that our approach achieves superior image quality with the same NFE budget. While [2] shows lower FID and CMMD values, they generate images with low prompt fidelity and degraded details indicated by lower CS and IR values. Please refer to Appendix B of the paper for further discussion of this phenomenon.
>
> On the other hand, we noticed that [1] is not applicable to our scenario. As observed in [4], the core assumption of monotonically increasing cosine similarity does not hold on most open-source diffusion model checkpoints.
>
> (The three parameters of THG corresponds to $(\rho, b, t_{hi})$.)
>
> | Variant                              | NFE | FID ↓   | CMMD ↓   | CS ↑     | IR ↑      |
> |--------------------------------------|-----|---------|----------|----------|-----------|
> | Selective Guidance [2]               | 74  | **13.394**  | **0.57148**  | 25.824   | 0.00972   |
> | Guidance Interval $(i>8, \omega=16)$ [3]  | 74  | 14.762  | 0.63085  | **26.261**   | **0.13308**   |
> | THG (1.0, 1.1, 39)                   | 74  | 14.156  | 0.59068  | 26.213   | 0.12260   |
> | | | | | | |
> | Selective Guidance [2]               | 70  | **12.895**  | **0.56052**  | 25.602   | -0.06691  |
> | Guidance Interval $(i>8, \omega=16)$ [3]  | 70  | 14.555  | 0.62227  | 26.153   | 0.09658   |
> | THG (1.1, 1.1, 38)                   | 70  | 14.232  | 0.59354  | **26.197**   | **0.11576**   |
> | | | | | | |
> | Selective Guidance [2]               | 66  | **12.180**  | **0.55062**  | 25.299   | -0.17810  |
> | Guidance Interval $(i>8, \omega=16)$ [3]  | 66  | 14.152  | 0.60939  | 25.998   | 0.04062   |
> | THG (1.3, 1.1, 35)                   | 66  | 14.243  | 0.59437  | **26.163**   | **0.09979**   |
>
> ---
>
> > * Gradually increasing the guidance scale by $b$ is kind of weird, as it breaks the fair comparison with baseline where the guidance scale is fixed.
>
> [A2] We thank the reviewer for this perceptive comment. We agree that using $b>1$ can appear to compromise a fair comparison with a fixed‑scale baseline. To address this, our ablation study (Tab. 2 of our paper) includes the $b=1$ case, which yields comparable CS and IR alongside superior FID and CMMD relative to other settings. Notably, even with a constant guidance scale $(b=1)$, our method continues to deliver strong performance, underscoring its effectiveness without any adjustment to the guidance weight.
>
> | Method                         | NFE ↓ | FID ↓   | CMMD ↓   | CS ↑    | IR ↑     |
> |---------------------|-------|---------|----------|---------|----------|
> | CFG (baseline)      | 100   | 14.057  | 0.58885  | 26.294  | 0.14765  |
> | | | | | | |
> | CFG-Cache w/o FFT   | 70    | 14.240  | 0.59187  | 26.141  | 0.08757  |
> | CFG-Cache           | 70    | 14.367  | 0.59556  | 26.180  | 0.09735  |
> | THG ($b=1.0$)       | 70    | **13.811**  | **0.58364**  | 26.137  | 0.09395  |
> | THG ($b=1.1$)       | 70    | 14.232  | 0.59354  | **26.197**  | **0.11576**  |
>
>
> ---
>
> > * The experiment could be conducted more thoroughly. Currently, each model only has one choice of NFE. Including more choices like 20 or 10 steps could further enrich the results and demonstrate the superiority of THG.
>
> [A3] We sincerely thank the reviewer for this valuable advice. We evaluated our method across multiple NFE settings to provide a more comprehensive comparison. Since the design of CFG-Cache requires a minimum NFE of 68, for fair comparison we report results with NFE ≥ 68. THG continues to give superior results, standing out in low NFE values. We will include additional results with more limited NFE settings in the final manuscript to clearly demonstrate the robustness and superiority of our approach.
>
> | Variant                  | NFE | FID ↓   | CMMD ↓   | CS ↑    | IR ↑     |
> |--------------------------|-----|---------|----------|---------|----------|
> | CFG baseline             | 100 | 14.133  | 0.58948  | 26.295  | 0.14764  |
> | | | | | | |
> | CFG-Cache w/o FFT        | 74  |  **14.145**  | 0.59163  | 26.238  | 0.11815  |
> | CFG-Cache                | 74  | 14.246  | 0.59497  | **26.272**  | **0.12979**  |
> | THG (1.0, 1.1, 39)       | 74  | 14.156  | **0.59068**  | 26.213  | 0.12260  |
> | | | | | | |
> | CFG-Cache w/o FFT        | 72  | **14.190** | **0.59282** | 26.206 | 0.10623 |
> | CFG-Cache                | 72  | 14.326 | 0.59604 | **26.240** | 0.11758 |
> | THG (1.05, 1.1, 39)      | 72  | 14.220 | **0.59282** | 26.213 | **0.11901** |
> | | | | | | |
> | CFG-Cache w/o FFT        | 70  | 14.282  | **0.59354**  | 26.138  | 0.08697  |
> | CFG-Cache                | 70  | 14.422  | 0.59759  | 26.179  | 0.09705  |
> | THG (1.1, 1.1, 38)       | 70  | **14.232**  | **0.59354**  | **26.197**  | **0.11576**  |
> | | | | | | |
> | CFG-Cache w/o FFT        | 68  | 14.418  | 0.60427  | 25.843  | –0.02967 |
> | CFG-Cache                | 68  | 14.678  | 0.61190  | 25.874  | –0.02859 |
> | THG (1.2, 1.1, 36)       | 68  | **14.303**  | **0.59258**  | **26.187**  | **0.10465**  |
>
> ---
>
> > * Though choosing the coarse time schedule based on error bound analysis is convincing, it takes considerable efforts to estimate the error constants. How is the performance gain of this carefully designed schedule? Could a naive choice also provide good performance? A naive choice of the coarse time schedule can also make the use of higher-order solvers reasonable.
>
> [A4] Thank you for your constructive suggestion. We evaluated THG with a couple of naive choices of the coarse grid $C$, namely a random one and an evenly spaced one. The results show the superiority of a carefully designed $C$.
>
> | Variant                | NFE | FID ↓   | CMMD ↓   | CS ↑    | IR ↑     |
> |------------------------|-----|---------|----------|---------|----------|
> | Random                 | 70  | 14.662  | 0.59890  | 26.167  | 0.07561  |
> | Evenly spaced          | 70  | 14.300  | **0.59068**  | 26.101  | 0.09282  |
> | $\rho=1.1$             | 70  | **14.232**  | 0.59354  | **26.197**  | **0.11576**  |
>
> ---
>
> > * It could be better to include the latency comparison with existing method to show the superiority of the proposed method.
>
> [A5] We thank the reviewer for this suggestion. We measured end‑to‑end inference latency of image generation on a single NVIDIA A100 GPU using bfloat16 precision. As expected, latency scales almost linearly with NFE since neural network evaluation dominates runtime. These results confirm that THG delivers improved image fidelity without sacrificing inference speed.
>
> | Variant                  | NFE | Latency (ms) |
> |--------------------------|-----|------------|
> | CFG baseline             | 100 |  746 |
> | | | | | | |
> | CFG-Cache w/o FFT        | 74  |  564 |
> | CFG-Cache                | 74  |  565 |
> | THG (1.0, 1.1, 39)       | 74  |  564 |
> | | | | | | |
> | CFG-Cache w/o FFT        | 72  |  550 |
> | CFG-Cache                | 72  |  551 |
> | THG (1.05, 1.1, 39)      | 72  |  550 |
> | | | | | | |
> | CFG-Cache w/o FFT        | 70  |  535 |
> | CFG-Cache                | 70  |  538 |
> | THG (1.1, 1.1, 38)       | 70  |  537 |
> | | | | | | |
> | CFG-Cache w/o FFT        | 68  |  522 |
> | CFG-Cache                | 68  |  522 |
> | THG (1.2, 1.1, 36)       | 68  |  521 |
>
> **References**
>
> [1] Castillo et al., "Adaptive guidance: Training-free acceleration of conditional diffusion models," arXiv preprint arXiv:2312.12487, 2023.
>
> [2] Golnari et al., "Selective guidance: Are all the denoising steps of guided diffusion important?," arXiv preprint arXiv:2305.09847, 2023.
>
> [3] Kynkäänniemi et al., "Applying guidance in a limited interval improves sample and distribution quality in diffusion models," NeurIPS 2024.
>
> [4] Zhang et al., "How Much To Guide: Revisiting Adaptive Guidance in Classifier-Free Guidance Text-to-Vision Diffusion Models," arXiv preprint arXiv:2506.08351, 2025.

---

> > ### Comment · Reviewer_FF6M · 2025-08-04
> >
> > I thank the authors for the detailed answers to my questions. Most of my concerns are addressed. I have a final question that how is the performance of the CFG baseline with NFE of 70 in Table 1 and 38 in Table 2?

---

> > > ### Author Response · Authors · 2025-08-07
> > >
> > > Dear reviewer FF6M,
> > >
> > > **tl;dr:**
> > > - For SD1.5, standard CFG (N=35, NFE=70) was superior over not only THG (N=50, NFE=70) but also CFG (N=50, NFE=100).
> > > - We found PSNR w.r.t. CFG (N=50, NFE=100) drops to 19.42 dB for CFG (N=35, NFE=70), compared to 24.16 dB of THG (N=50, NFE=70), indicating that a smaller N yields structurally different outputs.
> > > - As shown in [3, 4], FID can be unreliable when comparing image sets with different structures, so we should avoid direct head-to-head comparisons under mismatched N.
> > > - When matched for N, THG outperforms other NFE-reduction techniques.
> > >
> > > ---
> > >
> > > Thank you for your feedback. We compared our method against the CFG baseline. Note that we have to use a different number of fine-grained timesteps (i.e., N) to control the NFE of CFG.
> > >
> > > (The three parameters of THG corresponds to $(\rho, b, t_{hi})$.)
> > >
> > > | Variant (SD 1.5)        | N  | NFE | FID ↓  | CMMD ↓  | CS ↑   | IR ↑     |
> > > | ----------------------- | -- | --- | ------ | ------- | ------ | -------- |
> > > | CFG                     | 50 | 100 | 14.133 | 0.58948 | 26.295 | 0.14764  |
> > > | Selective Guidance [1]  | 50 | 70  | **12.895** | **0.56052** | 25.602 | -0.06691 |
> > > | Guidance Interval [2]   | 50 | 70  | 14.555 | 0.62227 | 26.153 | 0.09658  |
> > > | CFG-Cache [5]           | 50 | 70  | 14.422 | 0.59759 | 26.179 | 0.09705  |
> > > | THG (1.1, 1.1, 38)      | 50 | 70  | 14.232 | 0.59354 | **26.197** | **0.11576**  |
> > > |                         |    |     |        |         |        |          |
> > > | CFG                     | 35 | 70  | 13.342 | 0.57232 | 26.321 | 0.12694  |
> > > | Selective Guidance [1]  | 35 | 49  | **12.299** | **0.54550** | 25.623 | -0.09007 |
> > > | Guidance Interval [2]   | 35 | 49  | 14.490 | 0.61178 | 26.140 | 0.05919  |
> > > | CFG-Cache [5]           | 35 | 49  | 13.773 | 0.58162 | 26.120 | 0.05246  |
> > > | THG (1.7, 1.1, 30)      | 35 | 49  | 13.468 | 0.57637 | **26.265** | **0.09202**  |
> > > |                         |    |     |        |         |        |          |
> > > | CFG                     | 20 | 40  | 13.366 | 0.56159 | 26.370 | 0.10090  |
> > > | Selective Guidance [1]  | 20 | 30  | **12.839** | **0.54740** | 25.900 | -0.04452 |
> > > | Guidance Interval [2]   | 20 | 30  | 14.937 | 0.61333 | 26.244 | 0.02827  |
> > > | CFG-Cache [5]           | 20 | 30  | 13.675 | 0.56815 | 26.211 | 0.04604  |
> > > | THG (3.5, 1.1, 17)      | 20 | 30  | 13.345 | 0.56719 | **26.278** | **0.07212**  |
> > >
> > > For SD 1.5, although CFG (N=35, NFE=70) yields an FID of 13.342 (better than THG's 14.232 at N=50, NFE=70), its PSNR drops substantially to 19.42 dB compared to 24.16 dB for THG (N=50, NFE=70). This aligns with observations in [6], where reducing N sometimes lowers FID. Moreover, [3] and [4] show that FID may not reliably reflect perceptual quality when the image structures diverge, so we focus comparisons at the same N.
> > >
> > > CFG remains a strong reference point—using two NFEs per denoising step for guidance—but several works [1, 2, 5] have sought to improve efficiency by reducing NFEs. Under equivalent N settings, THG is superior to these baselines. We will include this comprehensive evaluation in the final manuscript to ensure a clear and fair comparison.
> > >
> > > We also provide results for SD 3.5 Large. THG shows better FID and IR compared to previous works.
> > >
> > > | Variant (SD 3.5 Large) | N  | NFE | FID ↓   | CMMD ↓   | CS ↑    | IR ↑     |
> > > | -- | -- | -- | -- | -- | -- | -- |
> > > | CFG | 28 | 56  | 68.149  | 0.81241  | 26.618  | 1.03034 |
> > > | CFG-Cache w/o FFT      | 28 | 38  | 69.330  | **0.71334**  | **26.692**  | 0.87915  |
> > > | CFG-Cache              | 28 | 38  | 69.715  | 0.72193  | 26.688  | 0.86095  |
> > > | THG (1.0, 1.2, 21)     | 28 | 38  | **68.732**  | 0.79619  | 26.652  | **1.01902**  |
> > > | | | | | | | |
> > > | CFG | 19 | 38  | 68.879  | 0.78320  | 26.694  | 1.00371 |
> > > | CFG-Cache w/o FFT      | 19 | 26  | 70.221  | **0.72454**  | 26.582  | 0.77238  |
> > > | CFG-Cache              | 19 | 26  | 70.911  | 0.75316  | 26.499  | 0.73825  |
> > > | THG (2.0, 1.2, 14)     | 19 | 26  | **68.898**  | 0.78475  | **26.614**  | **0.97243**  |
> > >
> > > Thank you again for your valuable questions. We believe these additions address your concern and strengthen our evaluation.
> > >
> > > ---
> > >
> > > **References**
> > >
> > > [1] Golnari et al., "Selective guidance: Are all the denoising steps of guided diffusion important?," arXiv preprint arXiv:2305.09847, 2023.
> > >
> > > [2] Kynkäänniemi et al., "Applying guidance in a limited interval improves sample and distribution quality in diffusion models," NeurIPS 2024.
> > >
> > > [3] Parmar et al., "On Aliased Resizing and Surprising Subtleties in GAN Evaluation," arXiv preprint arXiv:2104.11222, 2021.
> > >
> > > [4] Jung et al., "Internalized Biases in Fréchet Inception Distance," NeurIPS 2021 Workshop on Distribution Shifts, 2021.
> > >
> > > [5] Lv et al., "FasterCache: Training-Free Video Diffusion Model Acceleration with High Quality," arXiv preprint arXiv:2410.19355, 2024.
> > >
> > > [6] Choi et al., "Enhanced Diffusion Sampling via Extrapolation with Multiple ODE Solutions," ICLR 2025.

---

> > > > ### Comment · Reviewer_FF6M · 2025-08-07
> > > >
> > > > I thank the authors for the further results. I will update my final rating accordingly.

---

> > > > > ### Author Response · Authors · 2025-08-08
> > > > >
> > > > > Dear reviewer FF6M,
> > > > >
> > > > > Thank you for your follow-up and for considering our additional results. We appreciate the time you have devoted to our work during the rebuttal process. We will incorporate the newly presented results and clarifications into the final manuscript.

---

### Official Review · Reviewer_huS5 · 2025-07-02

**Clarity:** 3
**Significance:** 3
**Originality:** 3
**Rating:** 4
**Confidence:** 5

**Summary:**

This paper introduces Tortoise and Hare Guidance (THG), a training-free method to accelerate CFG sampling by reformulating it as a multirate system of ODEs. The key insight is that the noise estimate and additional guidance term have different sensitivities to numerical error. This allows the guidance term to be computed on a coarser timestep grid while maintaining the noise estimate on a fine grid. The method includes an error-bound analysis to determine appropriate coarse grid spacing and achieves up to 30% reduction in NFEs while preserving generation quality on text-to-image generation.

**Questions:**

Algorithm 3 appears to compute the coarse grid for each example individually, which seems correct from an accuracy perspective. However, I'm curious about the variance in coarse time grids across different initial noise samples and text prompts. How much do these grids differ? For practical efficiency, would it be possible to compute coarse grids in a batch-wise manner across multiple text prompts?


As I found this work interesting, If the authors could provide sufficiently convincing explanations or additional experiments that address the experimental concerns I raised, I would be willing to increase my score.

**Ethical Concerns:**

["NO or VERY MINOR ethics concerns only"]

**Final Justification:**

While the experimental validation appears somewhat limited, the paper proposes a novel approach and the rebuttal has largely addressed my concerns.

**Limitations:**

yes.

**Quality:**

2

**Strengths And Weaknesses:**

**Strength**

The ODE formulation of CFG and decomposition of CFG into separate tortoise and hare equations are quite novel. Especially, the decomposition of CFG based on their different sensitivities seems well-supported by the empirical evidence shown in Figure 2. The idea of treating these components differently is intuitive and their algorithms are mathematically sound. The error-bound analysis makes their approach more grounded approach.

**Weakness**
1. If I understand correctly, Algorithm 3 is used to obtain the coarse grid before running Algorithm 1. This preprocessing step likely requires additional NFEs that should be included when comparing computational efficiency. It would be more fair to account for these extra NFEs in the overall NFE comparison.

2. The paper only evaluates performance at specific NFE values. It would be valuable to see performance curves across a wider range of NFE values, particularly at more limited NFE settings, to better understand when and how much the method helps.

3. The experiments are limited to first-order solvers (DDIM, Euler). Since higher-order solvers like DPM-Solver are commonly used in practice, demonstrating the effectiveness on these methods would strengthen the paper's impact.

4. Given that four different evaluation metrics are used (FID, CMMD, CS, IR), it would be helpful to include brief descriptions of these metrics in the experimental section or appendix for readers who may not be familiar with all of them.


minor:
The notation for the difference estimation could be clearer to avoid confusion, as the current notation might be interpreted as a product of delta and epsilon.

---

> ### Author Rebuttal · Authors · 2025-07-31
>
> Dear reviewer huS5,
>
> Thank you for the constructive feedback and insightful questions. Your comments on the novelty and soundness of THG are encouraging. Below, we address your concerns.
>
> > 1. If I understand correctly, Algorithm 3 is used to obtain the coarse grid before running Algorithm 1. This preprocessing step likely requires additional NFEs that should be included when comparing computational efficiency. It would be more fair to account for these extra NFEs in the overall NFE comparison.
>
> [A1] We sincerely thank the reviewer for pointing out the potential ambiguity in our description of Algorithm 3 and apologize for any confusion this may have caused. In the final manuscript, we will clarify that:
>
> 1. **One-time preprocessing** : Algorithm 3 is executed *once* offline to compute that coarse grid $C$, incurring an $O(1)$ cost that is negligible compared to the $O(N)$ cost of processing $N$ requests.
> 2. **Reusable grid** : Once computed, $C$ can be reused for all subsequent inferences without any additional NFEs as long as the model and the domain is not changed.
> 3. **Flexible sampling**: It is also possible to compute $C$ using fewer sample runs. To demonstrate this, we estimated $C$ using a batch of 1,000 samples and found that it closely matched our original, large-scale estimate with 30,000 samples as the following. Compared to the original estimate, the 1,000 sample estimate demonstrated 95.25\% IoU (i.e., Jaccard index) in average over 30 trials.
>
> | Samples     | IoU | Steps |
> |-------------|-----|----------------------------------------------------------------------------------------------------------------------------------------------|
> | 30k (original) | - | 0, 1, 2, 3, 4, 5, 6, 8, 10, 12, 14, 17, 20, 23, 26, 28, 30, 32, 34, 36, 38, 39, 40, 41, 42, 43, 44, 45, 46, 47, 48, 49, 50 |
> | | | |
> | 1k (trial 1)      | 100\% | 0, 1, 2, 3, 4, 5, 6, 8, 10, 12, 14, 17, 20, 23, 26, 28, 30, 32, 34, 36, 38, 39, 40, 41, 42, 43, 44, 45, 46, 47, 48, 49, 50 |
> | 1k (trial 2)      | 100\% | 0, 1, 2, 3, 4, 5, 6, 8, 10, 12, 14, 17, 20, 23, 26, 28, 30, 32, 34, 36, 38, 39, 40, 41, 42, 43, 44, 45, 46, 47, 48, 49, 50 |
> | 1k (trial 3)      | 100\% | 0, 1, 2, 3, 4, 5, 6, 8, 10, 12, 14, 17, 20, 23, 26, 28, 30, 32, 34, 36, 38, 39, 40, 41, 42, 43, 44, 45, 46, 47, 48, 49, 50 |
> | ... | ... | ... |
> | 1k (trial 8)      | 57.5\% |0, 1, 2, 3, 4, 5, 7, 9, 11, 13, 16, 19, 22, 25, 28, 30, 32, 34, 36, 38, 39, 40, 41, 42, 43, 44, 45, 46, 47, 48, 49, 50 |
> | ... | ... | ... |
> | 1k (trial 30)      | 100\% | 0, 1, 2, 3, 4, 5, 6, 8, 10, 12, 14, 17, 20, 23, 26, 28, 30, 32, 34, 36, 38, 39, 40, 41, 42, 43, 44, 45, 46, 47, 48, 49, 50 |
>
> ---
>
> > 2. The paper only evaluates performance at specific NFE values. It would be valuable to see performance curves across a wider range of NFE values, particularly at more limited NFE settings, to better understand when and how much the method helps.
>
> [A2] Thank you for the constructive advice. Below, we report our method’s performance evaluated at various NFE values. Since the design of CFG-Cache requires a minimum NFE of 68, for fair comparison we report results with NFE ≥ 68. In the final manuscript, we will include comprehensive performance curves plotting key quality metrics (e.g., FID, CLIP score) across a wide spectrum of more limited NFE settings. (The three parameters of THG corresponds to $(\rho, b, t_{hi})$.)
>
> | Variant                  | NFE | FID ↓   | CMMD ↓   | CS ↑    | IR ↑     |
> |--------------------------|-----|---------|----------|---------|----------|
> | CFG baseline             | 100 | 14.133  | 0.58948  | 26.295  | 0.14764  |
> | | | | | | |
> | CFG-Cache w/o FFT        | 74  |  **14.145**  | 0.59163  | 26.238  | 0.11815  |
> | CFG-Cache                | 74  | 14.246  | 0.59497  | **26.272**  | **0.12979**  |
> | THG (1.0, 1.1, 39)       | 74  | 14.156  | **0.59068**  | 26.213  | 0.12260  |
> | | | | | | |
> | CFG-Cache w/o FFT        | 72  | **14.190** | **0.59282** | 26.206 | 0.10623 |
> | CFG-Cache                | 72  | 14.326 | 0.59604 | **26.240** | 0.11758 |
> | THG (1.05, 1.1, 39)      | 72  | 14.220 | **0.59282** | 26.213 | **0.11901** |
> | | | | | | |
> | CFG-Cache w/o FFT        | 70  | 14.282  | **0.59354**  | 26.138  | 0.08697  |
> | CFG-Cache                | 70  | 14.422  | 0.59759  | 26.179  | 0.09705  |
> | THG (1.1, 1.1, 38)       | 70  | **14.232**  | **0.59354**  | **26.197**  | **0.11576**  |
> | | | | | | |
> | CFG-Cache w/o FFT        | 68  | 14.418  | 0.60427  | 25.843  | –0.02967 |
> | CFG-Cache                | 68  | 14.678  | 0.61190  | 25.874  | –0.02859 |
> | THG (1.2, 1.1, 36)       | 68  | **14.303**  | **0.59258**  | **26.187**  | **0.10465**  |
>
> ---
>
> > 3. The experiments are limited to first-order solvers (DDIM, Euler). Since higher-order solvers like DPM-Solver are commonly used in practice, demonstrating the effectiveness on these methods would strengthen the paper's impact.
>
> [A3] We appreciate the reviewer for highlighting this point. We evaluated our approach with second-order solvers, namely DPM‑Solver‑2 and 2nd-order linear multistep method. Empirical results confirm that our method retains sample fidelity more effectively while offering similar efficiency improvements across different solver orders.
>
> | Variant                                       | NFE | FID ↓    | CMMD ↓   | CS ↑     | IR ↑     |
> |-----------------------------------------------|-----|----------|----------|----------|----------|
> | DPM-Solver-2                                  | 100 | 13.255   | 0.60379  | 26.254   | 0.16148  |
> | + CFG-cache w/o FFT                           | 70  | 13.387   | **0.60665**  | 26.107   | 0.10513  |
> | + CFG-cache                                   | 70  | 13.468   | 0.60880  | 26.147   | 0.11474  |
> | + THG                                         | 70  | **12.909**   | 0.60868  | **26.205**   | **0.14926**  |
> | | | | | | |
> | 2nd-order Linear Multistep Method             | 100 | 13.540   | 0.60653  | 26.260   | 0.15966  |
> | + CFG-cache w/o FFT                           | 70  | **13.686**   | **0.60844**  | 26.107   | 0.09881  |
> | + CFG-cache                                   | 70  | 13.798   | 0.61142  | 26.144   | 0.10805  |
> | + THG                                         | 70  | **13.686**   | 0.61094  | **26.204**   | **0.15184**  |
>
> ---
>
> > 4. Given that four different evaluation metrics are used (FID, CMMD, CS, IR), it would be helpful to include brief descriptions of these metrics in the experimental section or appendix for readers who may not be familiar with all of them.
>
> [A4] We thank the reviewer for pointing this out. To assist readers unfamiliar with these metrics, we will add concise definitions of FID, CMMD, CS, and IR in the final manuscript. For example:
>
> - **Metric**: To evaluate the methods, we use the following metrics: FID (Fréchet Inception Distance) compares the Inception‑v3 feature distributions of real and generated images; CMMD (CLIP Maximum Mean Discrepancy) quantifies the discrepancy between real and generated image distributions in the CLIP embedding space; CS (CLIP Score) is the average cosine similarity between each generated image and its corresponding text prompt in the joint CLIP embedding space; IR (ImageReward) denotes the mean reward assigned by a pretrained image‑text reward model to each generated sample.
>
> ---
>
> > minor: The notation for the difference estimation could be clearer to avoid confusion, as the current notation might be interpreted as a product of delta and epsilon.
>
> [A5] Thank you for your suggestion. While we have followed previous work [1] for the notation of $\delta \hat{\epsilon}\_c$, we will use $\Delta \hat{\epsilon}\_c$ or $\hat{\epsilon}\_{\delta}$ in the final manuscript to make the paper more understandable.
>
> ---
>
> > Algorithm 3 appears to compute the coarse grid for each example individually, which seems correct from an accuracy perspective. However, I'm curious about the variance in coarse time grids across different initial noise samples and text prompts. How much do these grids differ? For practical efficiency, would it be possible to compute coarse grids in a batch-wise manner across multiple text prompts?
>
> [A6] Thank you for this insightful question. It is certainly possible to compute the coarse grid in a batch-wise manner, and that is how we compute $C$ once on the dataset and utilize it afterwards in the paper. As shown in the table of [A1], we can estimate $C$ using a batch of 1,000 samples. This is accurate enough compared with the estimate on 30,000 samples, since the approximation error has low variance across different initial noise draws and text prompts (Fig. 3 of our paper). We will clarify that we compute $C$ in a batch-wise manner and use it for all samples in our final manuscript.
>
> **References**
>
> [1] Chung et al., "CFG++: Manifold-constrained Classifier Free Guidance for Diffusion Models," arXiv preprint arXiv:2406.08070, 2024.

---

> > ### Comment · Reviewer_huS5 · 2025-08-04
> >
> > I appreciate the author for the response, but my concerns about W2 remain.
> > The justification for comparing solely with CFG-Cache is still not provided, and the performance gap shown does not seem particularly compelling. To properly assess the contribution, I suggest a trend comparison against other baselines, especially standard CFG, across a wider range of NFEs. I think this is crucial for demonstrating a clear advantage.

---

> > > ### Author Response · Authors · 2025-08-07
> > >
> > > Dear reviewer huS5,
> > >
> > > **tl;dr:**
> > > - We extended our comparison to include additional baselines [1, 2] across a wider range of NFEs.
> > > - Standard CFG (N=35, NFE=70) was superior over not only THG (N=50, NFE=70) but also CFG (N=50, NFE=100).
> > > - We found PSNR w.r.t. CFG (N=50, NFE=100) drops to 19.42 dB for CFG (N=35, NFE=70), compared to 24.16 dB of THG (N=50, NFE=70), indicating that a smaller N yields structurally different outputs.
> > > - As shown in [3, 4], FID can be unreliable when comparing image sets with different structures, so we should avoid direct head-to-head comparisons under mismatched N.
> > > - When matched for N, THG outperforms other NFE-reduction techniques.
> > >
> > > ---
> > >
> > > Thank you for sharing your concerns. We compared our results with additional baselines such as [1] and [2]. Following your suggestion, we also compared our method against standard CFG across a wider range of NFEs. Note that we have to use a different number of fine-grained timesteps (i.e., N) to control the NFE of CFG.
> > >
> > > (The three parameters of THG corresponds to $(\rho, b, t_{hi})$.)
> > >
> > > | Variant                 | N  | NFE | FID ↓  | CMMD ↓  | CS ↑   | IR ↑     |
> > > | ----------------------- | -- | --- | ------ | ------- | ------ | -------- |
> > > | CFG                     | 50 | 100 | 14.133 | 0.58948 | 26.295 | 0.14764  |
> > > | Selective Guidance [1]  | 50 | 70  | **12.895** | **0.56052** | 25.602 | -0.06691 |
> > > | Guidance Interval [2]   | 50 | 70  | 14.555 | 0.62227 | 26.153 | 0.09658  |
> > > | CFG-Cache [5]           | 50 | 70  | 14.422 | 0.59759 | 26.179 | 0.09705  |
> > > | THG (1.1, 1.1, 38)      | 50 | 70  | 14.232 | 0.59354 | **26.197** | **0.11576**  |
> > > |                         |    |     |        |         |        |          |
> > > | CFG                     | 35 | 70  | 13.342 | 0.57232 | 26.321 | 0.12694  |
> > > | Selective Guidance [1]  | 35 | 49  | **12.299** | **0.54550** | 25.623 | -0.09007 |
> > > | Guidance Interval [2]   | 35 | 49  | 14.490 | 0.61178 | 26.140 | 0.05919  |
> > > | CFG-Cache [5]           | 35 | 49  | 13.773 | 0.58162 | 26.120 | 0.05246  |
> > > | THG (1.7, 1.1, 30)      | 35 | 49  | 13.468 | 0.57637 | **26.265** | **0.09202**  |
> > > |                         |    |     |        |         |        |          |
> > > | CFG                     | 20 | 40  | 13.366 | 0.56159 | 26.370 | 0.10090  |
> > > | Selective Guidance [1]  | 20 | 30  | **12.839** | **0.54740** | 25.900 | -0.04452 |
> > > | Guidance Interval [2]   | 20 | 30  | 14.937 | 0.61333 | 26.244 | 0.02827  |
> > > | CFG-Cache [5]           | 20 | 30  | 13.675 | 0.56815 | 26.211 | 0.04604  |
> > > | THG (3.5, 1.1, 17)      | 20 | 30  | 13.345 | 0.56719 | **26.278** | **0.07212**  |
> > >
> > > While [1] shows lower FID and CMMD values, it generates images with low prompt fidelity and degraded details indicated by lower CS and IR values. Please refer to Appendix B of the paper for further discussion of this phenomenon. Aside from [1], THG achieves superior overall performance.
> > >
> > > Although CFG (N=35, NFE=70) yields an FID of 13.342 (better than THG's 14.232 at N=50, NFE=70), its PSNR drops substantially to 19.42 dB compared to 24.16 dB for THG (N=50, NFE=70). This aligns with observations in [6], where reducing N sometimes lowers FID. Moreover, [3] and [4] show that FID may not reliably reflect perceptual quality when the image structures diverge, so we focus comparisons at the same N.
> > >
> > > CFG remains a strong reference point—using two NFEs per denoising step for guidance—but several works [1, 2, 5] have sought to improve efficiency by reducing NFEs. Under equivalent N settings, THG is superior to these baselines. We will include this comprehensive evaluation in the final manuscript to ensure a clear and fair comparison.
> > >
> > > Thank you again for your valuable suggestions. We believe these additions address your concern and strengthen our evaluation.
> > >
> > > ----
> > >
> > > **References**
> > >
> > > [1] Golnari et al., "Selective guidance: Are all the denoising steps of guided diffusion important?," arXiv preprint arXiv:2305.09847, 2023.
> > >
> > > [2] Kynkäänniemi et al., "Applying guidance in a limited interval improves sample and distribution quality in diffusion models," NeurIPS 2024.
> > >
> > > [3] Parmar et al., "On Aliased Resizing and Surprising Subtleties in GAN Evaluation," arXiv preprint arXiv:2104.11222, 2021.
> > >
> > > [4] Jung et al., "Internalized Biases in Fréchet Inception Distance," NeurIPS 2021 Workshop on Distribution Shifts, 2021.
> > >
> > > [5] Lv et al., "FasterCache: Training-Free Video Diffusion Model Acceleration with High Quality," arXiv preprint arXiv:2410.19355, 2024.
> > >
> > > [6] Choi et al., "Enhanced Diffusion Sampling via Extrapolation with Multiple ODE Solutions," ICLR 2025.

---

> > > > ### Comment · Reviewer_huS5 · 2025-08-07
> > > >
> > > > Thank you for the additional experiments. I am satisfied with the response given the limited time. As there are limitations to its improvement, I believe it needs to be supported by extensive evaluations. I hope the reviewer will reflect these results and discussion in the revised manuscript, and I have decided to raise the score.

---

> > > > > ### Author Response · Authors · 2025-08-08
> > > > >
> > > > > Dear reviewer huS5,
> > > > >
> > > > > Thank you for your constructive feedback. We will ensure that the final manuscript reflects both the new results and the discussion provided. We sincerely appreciate your thoughtful engagement throughout the review process.

---

### Official Review · Reviewer_M4bX · 2025-07-02

**Clarity:** 3
**Significance:** 3
**Originality:** 3
**Rating:** 5
**Confidence:** 3

**Summary:**

The paper presents a method called "Tortoise and Hare Guidance" which accelerates the diffusion sampling under CFG with a coarser timestep grid. It took a principled approach by analyzing the ODEs and developed a method based on theoretical analysis. The proposed method gave a 30% reduction in NFE while maintaining sample quality with SD on Coco dataset.

**Questions:**

See the (-) points in the strength and weakness section. It also contains questions

**Ethical Concerns:**

["NO or VERY MINOR ethics concerns only"]

**Quality:**

3

**Strengths And Weaknesses:**

(+) The paper took a first-principled numerical-analysis-inspired approach to improving the CFG schedule. The novelty is good in my opinion.

(+) The paper is training free. It can be plugged into existing trained models with minimal changes.

(+) The experimental results are relatively strong in my opinion for the method it tested and the dataset it tested on

(+) The writing and the flow of the paper is good

(-) Only SD series of models are tested, and it only works for 1st order solvers?

(-) Only tested on a limited datasets such as COCO. Not sure how well it generalizes to other datasets / models, and how sensitive the hyperparameters are across datasets

---

> ### Author Rebuttal · Authors · 2025-07-31
>
> Dear reviewer M4bX,
>
> Thank you for your valuable feedback and comments. We are grateful for your appreciation of our numerical-analysis-inspired approach on the denoising process. We hereby answer your concerns and questions.
>
> > (-) Only SD series of models are tested, and it only works for 1st order solvers?
>
> [A1] Thank you for sharing your concerns. In addition to DDIM and the Euler method, we evaluated our approach with second-order solvers, namely DPM‑Solver‑2 and 2nd-order linear multistep method. Our results show that our method is not restricted to first-order solvers, and that THG preserves sample quality more effectively while providing comparable efficiency gains across different integrator orders. Furthermore, experiments on audio diffusion models in [A2] demonstrate that our approach extends beyond the SD model family.
>
> | Variant                                       | NFE | FID ↓    | CMMD ↓   | CS ↑     | IR ↑     |
> |-----------------------------------------------|-----|----------|----------|----------|----------|
> | DPM-Solver-2                                  | 100 | 13.255   | 0.60379  | 26.254   | 0.16148  |
> | + CFG-cache w/o FFT                           | 70  | 13.387   | **0.60665**  | 26.107   | 0.10513  |
> | + CFG-cache                                   | 70  | 13.468   | 0.60880  | 26.147   | 0.11474  |
> | + THG                                         | 70  | **12.909**   | 0.60868  | **26.205**   | **0.14926**  |
> | | | | | | |
> | 2nd-order Linear Multistep Method             | 100 | 13.540   | 0.60653  | 26.260   | 0.15966  |
> | + CFG-cache w/o FFT                           | 70  | **13.686**   | **0.60844**  | 26.107   | 0.09881  |
> | + CFG-cache                                   | 70  | 13.798   | 0.61142  | 26.144   | 0.10805  |
> | + THG                                         | 70  | **13.686**   | 0.61094  | **26.204**   | **0.15184**  |
>
> ---
>
> > (-) Only tested on a limited datasets such as COCO. Not sure how well it generalizes to other datasets / models, and how sensitive the hyperparameters are across datasets
>
> [A2] Thank you for raising an important point. Exclusive use of the COCO dataset is still the standard for benchmarking T2I diffusion models [1-3] and ensures fair comparison with previous work. Nevertheless, we agree that validating across several benchmarks is important. Consequently, we conducted preliminary audio generation experiments with AudioLDM2 [4]. We used the validation set of AudioCaps [5] as the audio-text pair dataset and measured the performance with FAD [6] and CLAP Score [7], each corresponding to FID and CLIP Score. (The three parameters of THG corresponds to $(\rho, b, t_{hi})$.)
>
> | Variant | NFE | FAD ↓ | CLAP Score ↑ |
> |---|---|---|---|
> | CFG (baseline) | 100 | 2.596 | 0.2409 |
> | | | | |
> | CFG-Cache w/o FFT | 72 | 2.777 | 0.2312 |
> | THG (0.9, 1.15, 41) | 72 | **2.732** | **0.2362** |
> | | | | |
> | CFG-Cache w/o FFT | 70 | 2.901 | 0.2251 |
> | THG (0.9, 1.15, 39) | 70 | **2.764** | **0.2342** |
> | | | | |
> | CFG-Cache w/o FFT | 68 | 3.503 | 0.1972 |
> | THG (0.95, 1.15, 38) | 68 | **2.803** | **0.2328** |
>
> We will present these results in the appendix. While we adjusted the hyperparameters to obtain the same NFE, similar values can be used across different models and datasets. The consistent superior performance of our method underscores its broad applicability.
>
> **References**
>
> [1] Chung et al., "CFG++: Manifold-constrained Classifier Free Guidance for Diffusion Models," ICLR 2025.
>
> [2] Podell et al., "SDXL: Improving Latent Diffusion Models for High-Resolution Image Synthesis," arXiv preprint arXiv:2307.01952, 2023.
>
> [3] Esser et al., "Scaling Rectified Flow Transformers for High-Resolution Image Synthesis," arXiv preprint arXiv:2403.03206, 2024.
>
> [4] Liu et al., "AudioLDM 2: Learning Holistic Audio Generation with Self-supervised Pretraining," arXiv preprint arXiv:2308.05734, 2023.
>
> [5] Kim et al., "Audiocaps: Generating captions for audios in the wild," NAACL-HLT 2019.
>
> [6] Kilgour et al., "Fréchet Audio Distance: A Metric for Evaluating Music Enhancement Algorithms," arXiv preprint arXiv:1812.08466, 2018.
>
> [7] Elizalde et al., "CLAP: Learning Audio Concepts From Natural Language Supervision", arXiv preprint arXiv:2206.04769, 2022.

---

### Official Review · Reviewer_KWZf · 2025-07-03

**Clarity:** 4
**Significance:** 4
**Originality:** 4
**Rating:** 5
**Confidence:** 4

**Summary:**

The authors propose Tortoise and Hare guidance,  a training free framework for reducing the number of function evaluations during guided sampling in diffusion model. This is achieved by splitting the evaluations into a branch that is sensitive to change in time step (noise estimate) and a second branch which is relatively less sensitive to change in time step (guidance term). The signal at each time step is then calculated as a sum of the noise estimate (tortoise branch) and the guidance term (hare branch). The noise estimated is updated on a finer time scale whereas the guidance term is only update on a coarse time grid. The qualitative performance is shown to be close to standard diffusion and comparable quantitative performance is reported with respect to relevant baselines.

**Questions:**

1. What is the result if $x_t^H$ is only updates at the coarse time steps and its approximation for intermediate fine time steps not calculated ? In particular, instead of calculating $x_s^H$ for $s<t$, would it be possible to use $x_t^H$ for every $s$ between $x_t$ and $x_{t+1}$.

2. The authors mainly mention the use of DDIM and Euler method for evaluation. Is this approach also applicable to higher order methods for step reduction, and if not, what are the major considerations?

**Ethical Concerns:**

["NO or VERY MINOR ethics concerns only"]

**Final Justification:**

The authors rebuttal address most of the concerns raised, and the included results help strengthen the claims of the paper.

**Limitations:**

An adequate treatment of the limitations of the approach has been provided. In particular the authors acknowledge that the approach is only tested on first order solvers and on limited evaluation datasets.

**Paper Formatting Concerns:**

Paper adheres to NeurIPS paper formatting guidelines.

**Quality:**

3

**Strengths And Weaknesses:**

## Strengths
1. **Writing quality**: The paper is well written with attention to detail. The core motivation for the approach and the central idea of the paper is presented in a manner that is easy to understand and appreciate.

2. **Simple and elegant**: The proposed solution is relatively straightforward and explained in detail aiding in the ease of reproduction.

3. **Theoretically Grounded**: The motivation for the multi rate approach is theoretically grounded and error bound analysis has been provided to support the choice and design of certain hyper-parametes .

4. **Adaptive guidance scale**: This is an interesting trick to help maintain quality of samples depending on the resolution of coarse grid.

5. **Generality**: The proposed approach is a fundamental change to denoising process, allowing it to be used in any kind of diffusion model. Additionally, since it is a training free approach, it can be swapped into existing inference pipelines easily.

## Weaknesses
1. **Limited Qualitative results**: The main paper shows only 2 qualitative result for the difference in quality between the proposed approach and other baselines. Although the appendix includes some more. Demonstrating this on a wide variety of examples would be beneficial to understand and appreciate the effectiveness of the approach.

2. **Effect of schedule**: No ablation or analysis has been provided to indicate the effect of the timestep schedule during inference. Are the error bounds and performance guarantees the same for all time step schedules? Some discussion about this would be insightful

3. **Calculation of boosting term**: The authors ablate this term as well as motivate the need for it. But since the sensitivity changes across time steps, should the $b$ term be time dependent? Some discussion on this would be helpful.

4. **Evaluations**: Although the authors acknowledge this in the limitation section, the approach has been evaluated only on the COCO dataset. Since the algorithm is general enough, checking the performance on a wide variety of diffusion models would be beneficial. Although not a strict requirement, this could potentially also be evaluate on other modalities like video and audio.

5. **Coarse grid calculation**: The coarse grid is calculated based on the error bounds. Is this calculation independent of starting noise and the guidance weight?

---

> ### Author Rebuttal · Authors · 2025-07-31
>
> Dear reviewer KWZf,
>
> Thank you for your valuable feedback and detailed comments. We appreciate your recognition of the theoritically grounded motivation, the straightforward implementation, and the generality of our method. We address your concerns and questions in the response below.
>
> > 1. **Limited Qualitative results**: The main paper shows only 2 qualitative result for the difference in quality between the proposed approach and other baselines. Although the appendix includes some more. Demonstrating this on a wide variety of examples would be beneficial to understand and appreciate the effectiveness of the approach.
>
> [A1] We thank the reviewer for this constructive feedback. We agree that including more qualitative examples in the main paper is crucial for demonstrating how well our approach works and how it can be used in other situations. We will gladly add more qualitative results for the main paper in the final version with increased page limits.
>
> ---
>
> > 2. **Effect of schedule**: No ablation or analysis has been provided to indicate the effect of the timestep schedule during inference. Are the error bounds and performance guarantees the same for all time step schedules? Some discussion about this would be insightful.
>
> [A2] Indeed, it is possible to use a different timestep schedule for some models. [1] suggested that the 'trailing' strategy should be used to decide the timestep schedule for DDIM, while we have used the original 'leading' strategy in our paper. We tested THG with the fine timestep grid decided by the 'trailing' strategy. The results show THG is still effective under this new timestep schedule.
>
> | Variant   | NFE | FID ↓   | CMMD ↓   | CS ↑    | IR ↑     |
> |---|---|---|---|---|---|
> | CFG (baseline) | 100 | 14.746  | 0.60033  | 26.302  | 0.14847  |
> | | | | | | |
> | CFG-Cache w/o FFT | 70  | 14.923  | **0.60296**  | 26.137  | 0.08263  |
> | CFG-Cache | 70  | 15.019  | 0.60725  | 26.169  | 0.09253  |
> | THG (Ours) | 70  | **14.834**  | 0.60307  | **26.176**  | **0.11392**  |
>
> ---
>
> > 3. **Calculation of boosting term**: The authors ablate this term as well as motivate the need for it. But since the sensitivity changes across time steps, should the $b$ term be time dependent? Some discussion on this would be helpful.
>
> [A3] Thank you for your insightful suggestion. We performed an additional ablation study on SD1.5 with two time-dependent $b$ scenarios: (1) $b$ linearly grows from 1.0 at $t = T$, to 1.2 at $t = 0$. (2) $b$ linearly falls from 1.2 at $t = T$, to 1.0 at $t = 0$. The results indicate that there is no appreciable difference in generation quality between time-dependent $b$ and constant $b$. Our preliminary results were inconclusive, but we appreciate the suggestion and intend to explore it in future work by conducting a sensitivity-driven search for improved $b$ schedules.
>
> | Variant | NFE |  FID ↓   | CMMD ↓   | CS ↑    | IR ↑     |
> |---|----|---|---|---|---|
> | $b: 1.0 \mapsto 1.2$ | 70 | **14.047** | **0.59008** | 26.185  | 0.11211  |
> | $b: 1.2 \mapsto 1.0$ | 70 | 14.413 | 0.59568 | **26.201**  | **0.11938**  |
> | $b: 1.1 \mapsto 1.1$ | 70 | 14.232 | 0.59354 | 26.197 | 0.11576 |
>
> ---
>
> > 4. **Evaluations**: Although the authors acknowledge this in the limitation section, the approach has been evaluated only on the COCO dataset. Since the algorithm is general enough, checking the performance on a wide variety of diffusion models would be beneficial. Although not a strict requirement, this could potentially also be evaluate on other modalities like video and audio.
>
> [A4] We appreciate the reviewer for this constructive suggestion. Although exclusive use of the COCO dataset remains the standard for benchmarking T2I diffusion models [2-4], we agree that validating across modalities is important. Consequently, we performed preliminary audio generation experiments with AudioLDM2 [5]. We used the validation set of AudioCaps [6] as the audio-text pair dataset and measured the performance with FAD [7] and CLAP Score [8], each corresponding to FID and CLIP Score. (The three parameters of THG corresponds to $(\rho, b, t_{hi})$.)
>
> | Variant | NFE | FAD ↓ | CLAP Score ↑ |
> |---|---|---|---|
> | CFG (baseline) | 100 | 2.596 | 0.2409 |
> | | | | | | |
> | CFG-Cache w/o FFT | 72 | 2.777 | 0.2312 |
> | THG (0.9, 1.15, 41) | 72 | **2.732** | **0.2362** |
> | | | | | | |
> | CFG-Cache w/o FFT | 70 | 2.901 | 0.2251 |
> | THG (0.9, 1.15, 39) | 70 | **2.764** | **0.2342** |
> | | | | | | |
> | CFG-Cache w/o FFT | 68 | 3.503 | 0.1972 |
> | THG (0.95, 1.15, 38) | 68 | **2.803** | **0.2328** |
>
> We will include these results in the appendix. We believe that this expanded evaluation further demonstrates the generality and robustness of our approach.
>
> ---
>
> > 5. **Coarse grid calculation**: The coarse grid is calculated based on the error bounds. Is this calculation independent of starting noise and the guidance weight?
>
> [A5] We thank the reviewer for raising this important point. The error bounds are computed over the COCO validation set by sampling independent initial noise per prompt. Therefore, they characterize the full noise manifold and are independent to any single noise.
>
> For the guidance weight (or guidance scale) $\omega$, we have treated it as a fixed model parameter in our experiments. We expect that as $\omega$ increases, both the hare term and its approximation error grow, resulting in a denser coarse grid. To substantiate this, we evaluated the coarse timestep grid $C$ under several $\omega$ values.
>
> | Variant | Steps |
> |---|---|
> | $\omega=6.5, \rho=0.93$   | 0, 1, 2, 3, 4, 5, 6, 8, 10, 12, 14, 17, 20, 23, 26, 28, 30, 32, 34, 36, 38, 39, 40, 41, 42, 43, 44, 45, 46, 47, 48, 49, 50 |
> | $\omega=6.5, \rho=1.1$   | 0, 1, 2, 3, 4, 6, 8, 10, 13, 16, 19, 22, 25, 28, 31, 33, 35, 37, 39, 41, 43, 44, 45, 46, 47, 48, 49, 50 |
> | $\omega=7.5, \rho=1.1$   | 0, 1, 2, 3, 4, 5, 6, 8, 10, 12, 14, 17, 20, 23, 26, 28, 30, 32, 34, 36, 38, 39, 40, 41, 42, 43, 44, 45, 46, 47, 48, 49, 50  |
> | $\omega=8.5, \rho=1.1$   | 0, 1, 2, 3, 4, 5, 6, 7, 8, 10, 12, 14, 16, 18, 20, 22, 24, 26, 28, 30, 32, 34, 36, 37, 38, 39, 40, 41, 42, 43, 44, 45, 46, 47, 48, 49, 50 |
> | $\omega=8.5, \rho=1.22$   | 0, 1, 2, 3, 4, 5, 6, 8, 10, 12, 14, 17, 20, 23, 26, 28, 30, 32, 34, 36, 38, 39, 40, 41, 42, 43, 44, 45, 46, 47, 48, 49, 50   |
>
> While a bigger guidance scale results in a denser $C$, the overall trend is consistent; one can obtain the same $C$ by adjusting $\rho$.
>
> ---
>
> > 1. What is the result if $x_t^H$ is only updates at the coarse time steps and its approximation for intermediate fine time steps not calculated ? In particular, instead of calculating $x_s^H$ for $s<t$, would it be possible to use $x_t^H$ for every $s$ between $x_t$ and $x_{t+1}$.
>
> [A6] As you suggested, updating $x_t^H$ only at the coarse time steps and reusing its most recent value for intermediate fine steps correspond to an alternative multirate integration method [9]. We implemented this alternative scheme, where $x_t^H$ remains constant for $t\notin C$, and report the results. Our findings indicate that while this approach offers negligible computational savings, it yields a modest decrease in approximation fidelity compared to our fully interpolated method.
>
> | Variant  | NFE | FID ↓   | CMMD ↓   | CS ↑    | IR ↑     |
> |-----|----|-----|------|-----|-----|
> | Your suggestion | 70  | 14.351  | 0.59390  | 26.168  | 0.10067  |
> | Ours | 70  | **14.232**  | **0.59354**  | **26.197**  | **0.11576**  |
>
> ---
>
> > 2. The authors mainly mention the use of DDIM and Euler method for evaluation. Is this approach also applicable to higher order methods for step reduction, and if not, what are the major considerations?
>
> [A7] Thanks for your constructive feedback. In addition to DDIM and the Euler method, we evaluated our approach with second-order solvers, namely DPM‑Solver‑2 and 2nd-order linear multistep method. Our results show that our method is not restricted to first-order solvers. THG preserves sample quality more effectively while providing comparable efficiency gains across different integrator orders.
>
> | Variant | NFE | FID ↓    | CMMD ↓   | CS ↑     | IR ↑     |
> |-----|-----|-----|-----|------|------|
> | DPM-Solver-2  | 100 | 13.255   | 0.60379  | 26.254   | 0.16148  |
> | + CFG-cache w/o FFT  | 70  | 13.387   | **0.60665**  | 26.107   | 0.10513  |
> | + CFG-cache   | 70  | 13.468   | 0.60880  | 26.147   | 0.11474  |
> | + THG  | 70  | **12.909**   | 0.60868  | **26.205**   | **0.14926**  |
> | | | | | | |
> | 2nd-order Linear Multistep Method  | 100 | 13.540   | 0.60653  | 26.260   | 0.15966  |
> | + CFG-cache w/o FFT | 70  | **13.686**   | **0.60844**  | 26.107   | 0.09881  |
> | + CFG-cache | 70  | 13.798   | 0.61142  | 26.144   | 0.10805  |
> | + THG | 70  | **13.686**   | 0.61094  | **26.204**   | **0.15184**  |
>
> **References**
>
> [1] Lin et al., "Common Diffusion Noise Schedules and Sample Steps are Flawed," arXiv preprint arXiv:2305.08891, 2023.
>
> [2] Chung et al., "CFG++: Manifold-constrained Classifier Free Guidance for Diffusion Models," ICLR 2025.
>
> [3] Podell et al., "SDXL: Improving Latent Diffusion Models for High-Resolution Image Synthesis," arXiv preprint arXiv:2307.01952, 2023.
>
> [4] Esser et al., "Scaling Rectified Flow Transformers for High-Resolution Image Synthesis," arXiv preprint arXiv:2403.03206, 2024.
>
> [5] Liu et al., "AudioLDM 2: Learning Holistic Audio Generation with Self-supervised Pretraining," arXiv preprint arXiv:2308.05734, 2023.
>
> [6] Kim et al., "Audiocaps: Generating captions for audios in the wild," NAACL-HLT 2019.
>
> [7] Kilgour et al., "Fréchet Audio Distance: A Metric for Evaluating Music Enhancement Algorithms," arXiv preprint arXiv:1812.08466, 2018.
>
> [8] Elizalde et al., "CLAP: Learning Audio Concepts From Natural Language Supervision", arXiv preprint arXiv:2206.04769, 2022.
>
> [9] Adrian Sandu, "Multirate time integration: an overview", Aug 2023. Presentation at the 2023 Los Alamos Workshop on Time Integration for Multiphysics.

---

> > ### Comment · Reviewer_KWZf · 2025-08-06
> > **Response to rebuttal**
> >
> > I appreciate the authors for their detailed rebuttal and additional experiments. The added context addresses most of my initial concerns and helps strengthen the confidence in this paper. To that end I am increasing my score for this paper. The authors are encouraged to particularly  include the new results for different modalities and different solver variants since it does a great job of highlighting the generalizability of the proposed approach.

---

> > > ### Author Response · Authors · 2025-08-07
> > >
> > > Dear reviewer KWZf,
> > >
> > > Thank you for your thoughtful feedback. Your comments have been very helpful in improving our paper, and we appreciate the time and support given to the rebuttal process. We will add the new results for different modalities and solver variants in our final manuscript.

---

### Comment · Area_Chair_42BA · 2025-08-06
**Reminder: Author–Reviewer Discussion Closing Soon**

Dear Reviewers,

This is a gentle reminder that the Author–Reviewer Discussion phase ends within two days (by August 8). Please take a moment to read the authors’ rebuttal and engage in the discussion.

Your participation is important to ensure a fair and constructive review process. If you feel your concerns have been sufficiently addressed, you may also submit your Final Justification and update your rating early. Thank you for your contributions.

Best,

AC

---

### Note · Authors · 2025-08-12

We thank all reviewers and the AC for their thoughtful feedback and constructive suggestions throughout the review process. We have carefully addressed each concern, conducted additional experiments, and expanded our evaluation to strengthen the paper.

Mainly, we:

- [huS5, FF6M] Added comparisons with additional baselines such as Selective Guidance and Guidance Interval across a wider range of NFEs, clarifying that THG achieves superior overall performance.
- [KWZf, M4bX, huS5] Demonstrated generality across solvers, showing that THG is effective not only with first-order methods (DDIM, Euler) but also with higher-order integrators (DPM-Solver-2, 2nd-order Linear Multistep Method), preserving quality while reducing NFE.
- [KWZf, M4bX] Extended evaluation beyond COCO to audio generation with AudioLDM2 on AudioCaps, confirming THG's applicability across modalities with consistent improvements.
- [huS5] Clarified that coarse grid estimation (Algorithm 3) is a one-time, offline step with small overhead and low variance.
- [KWZf, FF6M] Investigated alternative multirate integration schemes, confirming our chosen design yields the best trade-off between quality and efficiency.

We will incorporate these results and clarifications into the final manuscript to ensure fair comparisons. We believe these updates more clearly demonstrate THG's practical utility as a training-free and general-purpose acceleration method for guided diffusion sampling.

We sincerely appreciate the reviewers' constructive engagement, which has helped us substantially improve the paper.

---

### Decision · Program_Chairs · 2025-09-17

**Decision:**

Accept (poster)

**Comment:**

This paper introduces Tortoise and Hare Guidance (THG) which accelerates diffusion inference by splitting the diffusion ODE into two branches. Reviewers are satisfied with the writing and mathematical soundness of this paper but raised issues about insufficient quantitative results and ablation studies. Most of the concerns are addressed by further experimental results provided during rebuttal. In the end, the reviewers all recommend accept. With these results added into the revised version of the current submission, I think this work will meet the quality threshold of NeurIPS conference and hence recommend acceptance.